# Learn to Achieve Out-of-the-Box Imitation Ability from Only One Demonstration

## Abstract

Imitation learning (IL) enables agents to mimic expert behaviors. Most previous IL techniques focus on precisely imitating one policy through mass demonstrations. However, in many applications, what humans require is the ability to perform various tasks directly through a few demonstrations of corresponding tasks, where *the agent would meet many unexpected changes when deployed*. In this scenario, the agent is expected to not only imitate the demonstration but also adapt to unforeseen environmental changes. This motivates us to propose a new topic called imitator learning (ItorL), which aims to derive an imitator module that can *on-the-fly* reconstruct the imitation policies based on very *limited* expert demonstrations for different unseen tasks, without any extra adjustment. In this work, we focus on imitator learning based on only one expert demonstration. To solve ItorL, we propose Demo-Attention Actor-Critic (DAAC), which integrates IL into a reinforcement-learning paradigm that can regularize policies' behaviors in unexpected situations. Besides, for autonomous imitation policy building, we design a demonstration-based attention architecture for imitator policy that can effectively output imitated actions by adaptively tracing the suitable states in demonstrations. We develop a new navigation benchmark and a robot environment for ItorL and show that DAAC outperforms previous imitation methods *with large margins* both on seen and unseen tasks.

## 1 Introduction

Humans can learn skills by imitating others. This has inspired researchers to propose imitation learning (IL), which enables intelligent agents to learn new tasks from demonstrations (Ng & Russell, 2000; Ross & Bagnell, 2010). Advanced IL techniques have made great progress in imitating behavior policies in complex tasks through mass demonstrations, without relying on reward signals (Garg et al., 2021; Kostrikov et al., 2020; Yin et al., 2022) as standard reinforcement learning (RL) does (Sutton & Barto, 2018). However, in many applications, what humans require is performing various tasks out of the box through very limited demonstrations of corresponding tasks, where there are many unexpected changes when deployed. In this scenario, the agent is expected to not only imitate the demonstration but also adapt to unforeseen environmental changes. For autonomous vehicles, we would like the vehicle to park in different parking lots directly (Ahn et al., 2022; Kümmerle et al., 2009) by presenting a human navigation trajectory, where the agent should handle the unexpected human being when imitating the parking trajectories; For robot manipulation, we aim for a robot arm to perform a variety of tasks directly (Dance et al., 2021; Yu et al., 2019) by just giving the corresponding correct operation demonstrations, where the agent should handle unexpected disturbances too.

Based on these observations, in this work, we propose a new topic called **Imitator Learning** (ItorL). In ItorL, we require the agent to accomplish various tasks that require the same intrinsic skills, e.g., a navigation agent to reach different targets in different terrains, and a robot-arm agent to perform various manipulation tasks. The aim of ItorL is to derive an imitator module that can reconstruct task-specific policies out of the box based on very limited corresponding expert demonstrations. More precisely, in ItorL, although we might have many pre-collected demonstrations and simulators for training, when deployed, the expert demonstrations are expensive, so the demonstrations for imitating should be *very limited*, leading a large number of states without referable expert actions for standard IL; Besides, for user experience, it should not have any additional adjustment phases in

the process of deployment, i.e., the agent should have the *out-of-the-box imitation ability*, i.e., it can reconstruct imitation policies with respect to the given demonstrations without further fine-tuning.

In this work, we focus on ItorL based on only one single expert demonstration and propose a practical solution for ItorL called *Demo-Attention Actor-Critic* (DAAC). To enable the agent to take reasonable actions in the states unvisited in demonstrations, we design an effective imitator reward and employ it into a context-based meta-RL framework (Rakelly et al., 2019) for imitation, where the imitator policy takes actions conditioned on demonstrations as the task context. The imitator policy interacts with the environment and maximizes the long-term imitator rewards on all tasks based on the corresponding demonstrations. Thanks to the trial-and-error learning mechanism of RL, the imitator policy can explore and optimize itself to generally follow expert demonstrations even when facing unexpected situations. However, just taking demonstrations as the context vector is inefficient in utilizing the full knowledge beyond the demonstration trajectories, as demonstrations not only tell the agent which task to accomplish but also the way to accomplish it. To efficiently build the imitation policy with respect to the given demonstrations, we propose a demonstration-based attention (DA) architecture for the imitator-policy network construction. Instead of taking demonstration as a free context vector, we utilize the attention mechanism (Vaswani et al., 2017) to stimulate the imitator policy to learn to accomplish tasks by tracing the states in demonstration trajectories. In particular, actions are taken based on the expert actions of the best-matching expert states, which is computed by the attention score between the current state and the states in demonstrations. We argue that DA implicitly regularizes the policy behavior by formalizing the data-processing pipeline with the attention mechanism so that it significantly improves the efficiency of learning to imitate from input demonstrations and the generalization ability to unseen demonstrations.

In the experiments, we build a demo-navigation benchmark for ItorL, which is a navigation task under different complex mazes without global map information. The results indicate that our proposed algorithm, DAAC, significantly outperforms existing baselines on both training performance and generalization to new demonstrations and new maps. We also deploy DAAC to more complex robotic manipulation tasks, where it maintains a clear advantage over baseline methods that struggle to achieve success in these challenging environments. Besides, we provide evidence that the proposed algorithm has the potential to achieve further performance improvements by scaling up either the dataset size or the number of parameters.

## 2 PROBLEM FORMULATION OF IMITATOR LEARNING

In this section, we first give notations, descriptions, and the formal definition of imitator learning (ItorL) in Sec. 2.1, then we discuss topic based on only one demonstration in Sec. 2.2.

### 2.1 IMITATOR LEARNING

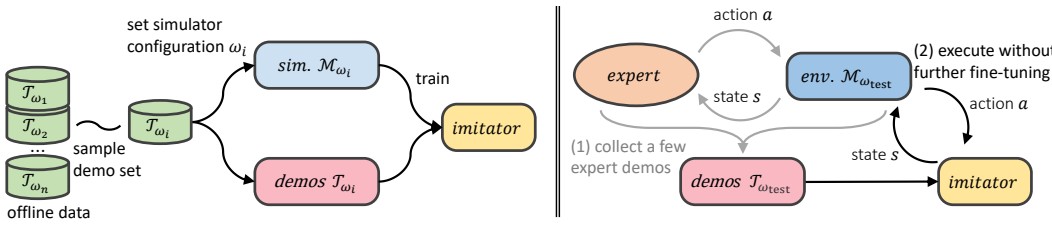

(a) train: learn a general model to imitate in all tasks      (b) deploy: adapt to the target task presented by a few demos

Figure 1: The paradigm of imitator learning. During the training process, an offline dataset with numerous expert demonstration sets $\{\mathcal{T}_{\omega_i}\}$ are provided, each of which can accomplish tasks $\mathcal{M}_\omega$ parameterized by $\omega$. The imitator policy is asked to reconstruct the expert policies for each task $\mathcal{M}_{\omega_i}$ based on the corresponding demonstrations $\mathcal{T}_{\omega_i}$. During deployment, experts interact in environments $\mathcal{M}_{\omega_\text{test}}$ and collect a few demonstrations $\mathcal{T}_{\omega_\text{test}}$ to mimic the experts without fine-tuning. Here we use "sim.", "env.", and "demos" as the abbreviation of simulator, environment, and demonstrations respectively.

In ItorL, we would like to derive a generalized imitator policy that can accomplish any unseen tasks through very limited expert demonstrations without further fine-tuning. For imitator policy training, we have pre-collected expert demonstrations from different tasks, along with the corresponding simulator for interacting. For imitator deployment, given any unseen task, we require the imitator

policy to use a few demonstrations to accomplish the task without further costly fine-tuning. Now we give the paradigm of ItorL in Fig. 1 and formal definition of ItorL in the following:

**Markov Decision Process**: We consider ItorL in a Markov Decision Process (MDP) (Sutton & Barto, 2018) $\mathcal{M}$ defined by a tuple $(\mathcal{S}, \mathcal{A}, T, R, d_0, \gamma)$, where $\mathcal{S}$ and $\mathcal{A}$ denote the state and action spaces, $T : \mathcal{S} \times \mathcal{A} \to P(\mathcal{S})$ describes a (stochastic) transition process, $R : \mathcal{S} \times \mathcal{A} \to \mathbb{R}$ is a bounded reward function, $d_0 \in P(\mathcal{S})$ is the initial state, and $\gamma \in (0, 1]$ denotes the discount factor. Here $P(X)$ denotes probability distributions over a set $X$. A policy $\pi : \mathcal{S} \to P(\mathcal{A})$ induces a Markov chain over the states based on $\mathcal{M}$. We use $\tau := \{s_0, a_0, \cdots, s_t, a_t\}$ to denote a trajectory, i.e., a sequence of state-action pairs for one episode of the Markov chain, where $s_i \in \mathcal{S}$ and $a_i \in \mathcal{A}$ are the state and action at timestep $i$.

**Task**: We formulate the concept "task" by parameterizing MDPs as $\mathcal{M}_\omega := (\mathcal{S}, \mathcal{A}, T_\omega, R_\omega, d_0, \gamma)$, where $\omega$ is the parameter of the MDP $\mathcal{M}_\omega$ in space $\Omega$. We assume that different MDPs share the same state and action spaces, initial state distribution, and discount factor. The difference on $T_\omega$ and $R_\omega$ can be defined by $\omega$.

**Reward Function $R_\omega$**: We only have the simplest reward function $R_\omega$ which can only indicate the ending of trajectories, e.g., $c$ for accomplishing the task, $0$ for failure, and $-c$ for dead.

**Unexpected Changes Modeling**: We formulate the unexpected changes from the period of demonstration collection to the agent execution into the stochasticity of $T_\omega$: between the two periods, the task parameters $\omega$ are shared, but the agent will reach unforeseen states because of the stochasticity. For example, in autonomous parking tasks, between the collection and execution periods, the agent is asked to park in the same parking lots (modeled by $\omega$), but pedestrians would occur randomly when the agent interacts with the environment (modeled by the stochasticity of $T_\omega$).

**Expert Demonstration**: We use $\tau_\omega$ to denote an expert demonstration that can accomplish the task in $\mathcal{M}_\omega$. Standard IL and their variant settings (Arora et al., 2020; Ross & Bagnell, 2010; Finn et al., 2017a;b; Li et al., 2021; Yu et al., 2018) do not assume the quality of the behavior policy to be imitated, and the reward function to complete the task $R_\omega$ is also unnecessary. These techniques are just asked to reconstruct any possible policies in the collected dataset. In ItorL, we require that the policy to conduct the demonstrations should be an expert that can complete the tasks defined by $R_\omega$. Specifically, we denote $\mathcal{T}_\omega := \{\tau_\omega^{(0)}, \tau_\omega^{(1)}, \cdots\}$ as an expert demonstration set in $\mathcal{M}_\omega$.

**Imitator Learning**: Now we formulate ItorL as follows. In ItorL, we would like to derive a generalized imitation policy $\Pi(a|s, \mathcal{T}_\omega)$ which can accomplish the task in $\mathcal{M}_\omega$ for any $\omega \in \Omega$, where $\mathcal{T}_\omega$ is an expert demonstration set for $\mathcal{M}_\omega$. For imitator policy training, we have pre-collected expert demonstrations $\{\mathcal{T}_\omega\}$ from different $\mathcal{M}_\omega$, along with the corresponding simulator of $\mathcal{M}_\omega$ for interacting. For imitator deployment, given any $\omega_{\text{test}} \in \Omega$, we require the imitator policy to use a few demonstrations $\mathcal{T}_{\omega_{\text{test}}}$ for $\Pi(a|s, \mathcal{T}_{\omega_{\text{test}}})$ to accomplish the task in $\mathcal{M}_{\omega_{\text{test}}}$ without further fine-tuning.

### 2.2 IMITATOR LEARNING BASED ON ONLY ONE DEMONSTRATION

In this work, we focus on ItorL based on a single demonstration. This section will formulate the conditions that make topic feasible based on a single demonstration.

A fundamental problem of ItorL is how can we use a single demonstration to reconstruct any expert policy, as it is inevitable that there *are a large number of states without referable expert actions* for imitation? Without further assumptions on the task-parameter space $\Omega$, it is easy to construct some ill-posed problems that it is impossible for a unified $\Pi(a|s, \mathcal{T}_\omega)$ to reconstruct all of the expert policies unless $\mathcal{T}$ covers the full state-action space. However, in many applications, it is unnecessary for $\Pi$ to imitate policies for any task. In the following, we give one practical task set $\mathbb{M} := \{\mathcal{M}_\omega \mid \omega \in \Omega\}$, that enables ItorL through *only one* demonstration.

**Definition 2.1** ($\tau_\Omega$-tracebackable MDP set). For an MDP set $\mathbb{M} := \{\mathcal{M}_\omega \mid \omega \in \Omega\}$, if there exists a unified goal-conditioned policy $\beta(a|s, g)$, $\forall \mathcal{M}_\omega \in \mathbb{M}$, for any $\tau_\omega$, we have $\forall s_i \in \tau_\omega$ or $\forall s_0 \in R(d_0)$, $\exists g_j \in \tau_\omega$, $\beta(a|s, g_j)$ can reach $g_j$ from $s = s_i$ within finite timesteps, where $R(X)$ is the state set in $X$, $i$ and $j$ denote the timestep of states in $\tau$ and $j > i$, then $\mathbb{M}$ is a $\tau_\Omega$-tracebackable MDP set.

**Proposition 2.2** (1-demo imitator availability). *If $\mathbb{M} := \{\mathcal{M}_\omega \mid \omega \in \Omega\}$ is a $\tau_\Omega$-tracebackable MDP set, there exists at least a unified imitator policy $\Pi(a|s, \mathcal{T}_\omega)$ that can accomplish any task in $\mathbb{M}$ only given one corresponding demonstration, i.e., $|\mathcal{T}_\omega| = 1$.*

The core in Prop. 2.2 is the unified goal-conditioned policy $\beta$ defined in Def. 2.1. The motivation behind $\beta$ is that, whatever the task we would like to imitate is, and whatever the unexpected changes in the environment will lead the agent to, the behaviors of coming back to the states in the demonstrations are general and consistent. The assumption is practical in many applications, for example, in the task of navigation for parking, we might meet unexpected obstacles and pedestrians in the processing of imitation, which don't exist in the demonstrations. However, for any parking lot, the behaviors to handle the situations are consistent: executing avoidance until the state is safe, then tracing back to the demonstration. If the policy $\beta$ exists, even the demonstration just gives us parts of the state-action pairs in the state-action space, we can imitate the demonstrations and reach the goal by repeatedly tracing a reachable successor state $g \in \tau_\omega$ and using $\beta$ to guide the agent until reaching the goal state. Similarly, for robot manipulation tasks, whatever disturbance a robot arm might encounter, if we always have a unified policy $\beta$ to reach some of the successor states in the demonstrations, we can reach the goal by repeatedly calling $\beta$ with suitable goals. Briefly note that it is unnecessary to ask for this consistent behavior for any states in the state space. As defined in Def. 2.1, the states in $\tau_\omega$ and $R(d_0)$ are enough for us to derive the 1-demo imitator availability, where the full derivation and discussion are in App. A.

However, so far, how to build the imitator policies from data is challenging, e.g., it is hard to make a goal-conditioned policy $\beta$ act through directly imitating $\tau_\omega$, and it is also complex to select suitable target states $g \in \tau_\omega$ to push forward the agent through $\beta$. In the next section, we will handle the above problem by interacting with the environment $\mathcal{M}_\omega$ for policy training.

## 3  RELATED WORK

We introduce Meta-IL, which is similar to ItorL in the following and leave the complete related work in Appendix, including IL (Sec. C.1), meta-IL(Sec. C.2), the combination of IL and RL (Sec. C.3), and context-based meta-RL (Sec. C.4). Meta-IL can be categorized into few-shot meta-IL and one-shot meta-IL: (1) Few-shot meta-IL aims to get a generalizable policy that can complete new tasks with only a few expert trajectories. The mainstream solutions utilize model-agnostic meta-learning (MAML) (Finn et al., 2017a) to learn initial task parameters and fine-tune them via a few steps of gradient descent to satisfy new task needs (Finn et al., 2017b; Li et al., 2021; Yu et al., 2018). However, these approaches need online interaction and extra computation infrastructure for gradient update and determining a suitable amount of fine-tuning steps before deployment (Finn et al., 2017a). ItorL is to create an imitator policy, $\Pi(a|s, \mathcal{T}_\omega)$, informed solely by a pre-collected expert demonstration set, **without requiring any fine-tuning**. During deployment, this policy simply takes in the relevant demonstration $\tau_\omega$ to generate the appropriate action for any given state. (2) One-shot meta-IL achieves generalizable imitation through context-based policy models (Dasari & Gupta, 2021; Duan et al., 2017; Mandi et al., 2022), such as Transformer (Vaswani et al., 2017), that take demonstrations as input. The core idea is to extract representations of demonstrations through these powerful fitting abilities of neural networks, and then use BC to reconstruct the imitation policy. However, the demonstrations for imitation are limited, the inevitable prediction errors on unseen states and the compounding errors of BC (Ross et al., 2011) hurt the capacities of these methods, especially in generalizing to new tasks (Mandi et al., 2022). Different from one-shot IL, in ItorL, the interactions with simulators of the demonstrations for training are allowed, and the demonstrations for imitation are assumed to come from experts. This allows us to stimulate the policy to imitate the experts and learn general behaviors to handle the situations unseen in the demonstrations via improving the performance in reward function, and finally enables us to **have the capacity to learn to imitate based on fewer demonstrations than the imitation algorithms in other settings.**

## 4  DEMO-ATTENTION ACTOR-CRITIC FOR IMITATOR LEARNING

In this section, we first introduce a basic context-based meta-RL framework adopted for solving ItorL in Sec. 4.1. To enable the agent to efficiently utilize the knowledge beyond the demonstrations, we give a novel network architecture for the actor and critic in Sec. 4.2. Finally, we integrate the meta-RL framework with the new network architecture to our final solution, which is in Sec. 4.3.

### 4.1  CONTEXT-BASED META-RL FRAMEWORK FOR IMITATOR LEARNING

Since the demonstrations are assumed to be performed by experts capable of accomplishing tasks defined by $R_\omega$, it is consistent between learning to improve the return defined by $R_\omega$ and imitation. On the other hand, *we can stimulate the imitator policy to imitate the target policies by improving the performance with $R_\omega$.* Along this line, we consider handling ItorL through context-based meta-RL

---

**Algorithm 1** Context-based Meta-RL framework for ItorL

---

**Input:** A task set $\mathbb{M}_{\text{train}}$, and a demonstration set $\{\mathcal{T}_{\omega_i}\}$ for each task $\mathcal{M}_{\omega_i} \in \mathbb{M}_{\text{train}}$
**Process:**
1: Initialize a task-information extractor $\phi$, context-based policy $\pi$, and a replay buffer $\mathcal{B}$
2: **for** $1, 2, 3, ...$ **do**
3:    Sample a task $\mathcal{M}_\omega$ from the sampling strategy $P(\mathbb{M}_{\text{train}})$
4:    Infer the demonstration representation $z = \phi(\mathcal{T}_\omega)$
5:    **for** $j = 1, 2, 3, ..., H$ **do**
6:       Sample an action $a_j \sim \pi(a|s_j, z)$
7:       Rollout one step $s_{j+1} \sim \mathcal{M}_\omega(s|s_j, a_j)$, get the reward $r_j = R_\omega(s_j, a_j)$
8:       Add $(s_j, a_j, r_j, s_{j+1}, \mathcal{T}_\omega)$ to $\mathcal{B}$
9:    **end for**
10:    Use SAC (Haarnoja et al., 2018) to update $\phi$ and $\pi$ with batch samples from $\mathcal{B}$
11: **end for**

---

techniques (Chen et al., 2021; OpenAI et al., 2019; Rakelly et al., 2019), where the pseudocode of the framework is in Alg. 1. In context-based meta-RL framework, the imitator policy $\Pi$ can be decomposed into a context-based policy $\pi$ and a task-information extractor $\phi$, i.e., $\Pi := \pi(a|s, \phi(\mathcal{T}_\omega))$. $\phi$ takes $\mathcal{T}_\omega$ as inputs, aiming to extract the representation of the task $\omega$ via latent variables $z \in \mathcal{Z}$. The context-based policy $\pi$ takes the states and the extracted latent variables as inputs, aiming to make adaptive decisions for each task. Specifically, for each task in $\mathcal{M}_\omega$, we infer the task presentation via $z = \phi(\mathcal{T})$, then infer the action via $a \sim \pi(a|s, z)$. A standard objective (Duan et al., 2017; OpenAI et al., 2019) for learning the optimal extractor $\phi^*$ and policy $\pi^*$ is:

$$\max_{\phi, \pi} \mathbb{E}_{\mathcal{M}_\omega \sim P(\mathbb{M}_{\text{train}})} \left[ \mathbb{E}_{\mathcal{M}_\omega, \phi, \pi} \left[ \sum_{i=0}^{\infty} \gamma^i R_\omega(s_i, a_i) \right] \right],$$

where $\mathbb{M}_{\text{train}}$ is the training task set, $P(\mathbb{M}_{\text{train}})$ is a sampling strategy for task $\mathcal{M}_\omega$ generating, and $\mathbb{E}_{\mathcal{M}_\omega, \phi, \pi}$ is the expectation over trajectory $\{s_0, a_0, s_1, a_1, ...\}$ sampled from $\mathcal{M}_\omega$ with $\phi$ and $\pi$. The context-aware policy $\pi$ is trained to take the optimal actions *in all the tasks sampled from* $P(\mathbb{M}_{\text{train}})$. The key to taking optimal actions in all tasks is that the parameters of $\phi$ will be updated through the policy gradients (Sutton & Barto, 2018) backpropagated from $\pi$. Thus, if the optimal actions are in conflict among different $\mathcal{M}$, the policy gradient will guide the extractor in distinguishing the representations among $\mathcal{T}$ until all the optimal actions under the inferred contexts have no conflict (Chen et al., 2021). Thus *if the task set $\mathbb{M}_{\text{train}}$ cover the task space $\Omega$, we can claim that, when deployed, the optimal policy $\Pi^* := \pi^*(a|s, \phi^*(\mathcal{T}_\omega))$ can take correct actions as in the training set.*

To generalize over unseen tasks, $\phi$ necessitates exposure to a sufficiently diverse task set $\mathbb{M}$ spanning the parameter space. However, it is almost impractical to construct a task set $\mathbb{M}_{\text{train}}$ to cover the task space $\Omega$. The generalization ability relies on the interpolation capabilities of neural networks. Previous studies also show that the behavior of $\phi$ to unseen tasks might be unstable without further constraints or regularization (Nagabandi et al., 2019; Wang et al., 2020). In the following, we will propose a new architecture for actors and critics to regularize the policy behavior.

### 4.2 DEMONSTRATION-BASED ATTENTION ARCHITECTURE

As mentioned before, the behavior of $\phi$ to unseen tasks might be unstable (Luo et al., 2022; Wang et al., 2020). Previous studies often handle the problem by adding extra losses/constraints to regularize the context representation (Dasari & Gupta, 2021). Besides, we also observe that just regarding demonstrations as context vectors are inefficient in fully mining the knowledge implied in these data efficiently, e.g., the demonstration sequence not only tells the agent which task to accomplish but the way to accomplish the task, finally hurting the efficiency of the algorithm to find the optimal $\Pi^*$. Based on the above observations, in this study, instead of utilizing auxiliary losses as in prior works, we implicitly constrain the "context representation" via the network architecture itself, i.e., the demonstration-based attention (DA) architecture. The architecture is based on the prior that, for any unobserved task in the $\tau_\Omega$-tracebackable MDP set, imitator actions can be taken in two general decision-making phases, which will be discussed below. The DA architecture stimulates the policy to make decisions following the general decision-making phases.

Inspired by Prop. 2.2, we build the DA architecture based on this intuition: *For imitation, the first step is to find a target state from the demonstration, which has high similarity with the current state. Then the second step is taking action based on the expert action corresponding to the target state.* In particular, utilizing the attention mechanism (Vaswani et al., 2017), DA uses the following two major phases to mimic the above process: (1) **Phase 1: determine the state to follow.** Attention weighting is a module in standard attention architecture (Vaswani et al., 2017), which outputs the similarity weights of the items in the key vector **k** compared with the query vector **q**. Specifically, one popular implementation is $\mathbf{w} = \mathrm{softmax}(\mathbf{q}\mathbf{k}^\top/\sqrt{d_k})$, where $d_k$ is the feature dimension of **k**, and $\mathbf{q}\mathbf{k}^\top$ is to compute the dot products of the query with the keys in all timesteps. The dot-product operation of **k** and **q** makes states with higher similarity output a larger attention weight. We utilize this architecture and let the representation of expert states be **k** and the visited state representation be **q**, to regularize the policy and determine the expert state to follow before decision-making; (2) **Phase 2: determine the action to take.** The attention weighting is followed by a point-wise multiplication to compute $v''$, i.e., $v'' = \sum_i v_i w_i$. Each value vector **v** is a presentation of the corresponding expert ac-

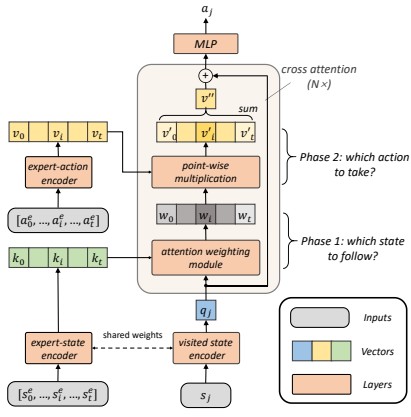

Figure 2: The DA architecture for the actor. $[s_0^e, ..., s_i^e, ..., s_t^e]$ denote expert states and $[a_0^e, ..., a_i^e, ..., a_t^e]$ the expert action list. $s_j$ is the visited state of the actor at timestep $j$. We use $q$, $k$, and $v$ to denote the query, key, and value vectors of an attention module. $N\times$ denotes an $N$-layer cross-atttention module, which takes the output $v''$ of the last layer as the input $q_j$ of the next layer.

tion. The point-wise multiplication applies the attention weight $w_i$ to the representation of action $a_i^e$ for each timestep $i$ to compute $a_j$. The critic is built with the same method, which is in App. D.

We use DA architecture to fulfill the roles of both $\phi$ and $\pi$ together to stimulate the policy to make decisions based on the discrepancy between the current state and the states in the demonstration. In a nutshell,*the regularizer in our context is essentially the inductive bias of the prior knowledge about the two-phase imitation introduced by the neural network architecture.* The above data-processing pipeline within the policy network implicitly guides the policy to take actions based on the expert action with attention weights so that it can improve the efficiency of learning to imitate from input demonstrations and the generalization ability to unseen demonstrations. We would like to point out a limitation that the DA architecture will also hurt the decision-making ability when the task set is not a $\tau_\Omega -$ tracebackable MDP set defined in Def. 2.1, i.e., it does not exist a unified goal-conditioned policy $\beta$ for solving ItorL in $\mathbb{M}$. For example, when the current state might be too distant from any expert state for some *inevitable* reasons, the attention mechanism would fail to match any state, degrading the architecture to mere guesswork. However, through our experiments, we found that to some degree the attention mechanism can still consolidate actions from several locally similar states of the expert to produce the correct action. The detailed discussion can be seen in App. F.

### 4.3 DEMO-ATTENTION ACTOR-CRITIC

We summarize our practical solution for ItorL as *Demo-Attention Actor-Critic* (DAAC). DAAC follows the context-based meta-RL framework in Alg. 1, where the imitator policy uses DA architecture as an integrated implementation of context-based policy $\pi$ and task-information extractor $\phi$.

Besides, for further regularizing the policy's behavior in states unvisited in demonstrations, we embed the imitation process to RL with a general stationary imitator reward derived from a single demonstration, which enables policy learning by imitating the input demonstration instead of from scratch by ending rewards. Inspired by Ciosek (2022), which has shown that IL can be done by RL with a constructed stationary reward, we heuristically design an ItorL reward $R_{\mathrm{Itor}}$ to embed the imitation process into the RL in a similar way. We leave the full discussion in App. B. In summary, we construct an imitator reward function:

$$R_{\mathrm{Itor}}(s,a) := 1 - \min\left\{ \underbrace{d(\bar{s},s)^2}_{\text{distance to state } \bar{s}} + \underbrace{d(\bar{a},a)^2/\exp(d(\bar{s},s)^2)}_{\text{weighted distance to action } \bar{a}}, \eta\right\} + \alpha R_\omega(s,a), \quad (1)$$

where $\bar{s}, \bar{a}$ is the nearest expert state-action pair: $(\bar{s},\bar{a}) = \arg\min_{(s',a')\in\mathcal{T}} d(s,s')^2$. The selected action $\bar{a}$ corresponds to the action associated with state $\bar{s}$ in the transition pair. $\eta$ is a hyperparameter

Table 1: Success rate comparisons on demo-navigation tasks. The agent needs to imitate demos seen during the training, new demos from seen maps, and demos collected on new maps, namely denoted as "seen", "new_demo", and "new_map" in this table. Our experiment uses 3 random seeds and we **bold** the best scores for each task.

| Map Type | Single-Map | | | | Multi-Map | | | | | |
|---|---|---|---|---|---|---|---|---|---|---|
| Obstacle Type | Non-Obstacle | | Obstacle | | Non-Obstacle | | | Obstacle | | |
| Demontrations | seen | new_demo | seen | new_demo | seen | new_demo | new_map | seen | new_demo | new_map |
| **Coord** DAAC | **1.00±0.00** | **0.94±0.03** | **0.81±0.02** | **0.76±0.02** | **0.92±0.02** | **0.87±0.04** | **0.86±0.02** | **0.77±0.03** | **0.77±0.03** | **0.73±0.02** |
| DCRL | 0.99±0.01 | 0.93±0.01 | 0.78±0.03 | 0.74±0.03 | 0.44±0.03 | 0.32±0.02 | 0.31±0.00 | 0.51±0.01 | 0.50±0.02 | 0.46±0.02 |
| TRANS-BC | 0.43±0.09 | 0.16±0.10 | 0.14±0.10 | 0.04±0.02 | 0.50±0.07 | 0.29±0.05 | 0.30±0.07 | 0.32±0.05 | 0.22±0.03 | 0.21±0.04 |
| CbMRL | 0.98±0.00 | 0.76±0.02 | 0.66±0.01 | 0.44±0.02 | 0.28±0.02 | 0.29±0.03 | 0.26±0.03 | 0.37±0.03 | 0.32±0.03 | 0.33±0.02 |
| **No-Coord** DAAC | **0.51±0.19** | **0.71±0.06** | **0.46±0.06** | **0.58±0.04** | **0.83±0.03** | **0.63±0.01** | **0.54±0.05** | **0.50±0.02** | **0.45±0.02** | **0.40±0.03** |
| DCRL | 0.24±0.03 | 0.01±0.01 | 0.15±0.01 | 0.00±0.00 | 0.15±0.06 | 0.05±0.02 | 0.04±0.02 | 0.11±0.02 | 0.03±0.02 | 0.05±0.02 |
| TRANS-BC | 0.06±0.02 | 0.00±0.00 | 0.02±0.02 | 0.00±0.00 | 0.06±0.04 | 0.01±0.01 | 0.02±0.01 | 0.02±0.03 | 0.03±0.02 | 0.01±0.01 |
| CbMRL | 0.15±0.06 | 0.02±0.01 | 0.09±0.02 | 0.01±0.00 | 0.14±0.01 | 0.03±0.01 | 0.02±0.01 | 0.10±0.02 | 0.05±0.02 | 0.06±0.01 |

that clips the distance penalty calculated based on the too-far state pairs into a fixed constant, and $\alpha$ is a rescale coefficient. $d(\cdot, \cdot)$ measures the distance between two inputs, and it can be customized differently for different tasks, which is L2 distance in this work. Finally, we take the standard soft actor-critic algorithm (Haarnoja et al., 2018) for policy learning in DAAC. More implementation details of DAAC are in App. D and the algorithm is listed in Alg. 2.

## 5 EXPERIMENT

In the experiment, we build a demo-navigation benchmark for ItorL, which is a navigation task under different complex mazes without global map information. We introduce this benchmark in Sec. 5.1 followed by our experiment setup in Sec. 5.2. In Sec. 5.3, we evaluate our method from various perspectives, including training performance, generalization ability to unseen demonstrations, and unexpected situations. We then verify the effects of the DA architecture and proposed imitator reward in Sec. 5.4. In Sec. 5.5, we show that the proposed algorithm has the potential to achieve further performance improvements by scaling up either the dataset size or the number of parameters. Finally, we provide experimental results on more complex tasks in Sec. 5.6.

### 5.1 BENCHMARK FOR IMITATOR ABILITY IN UNSEEN SITUATIONS

We use a simple environment to construct a *challenging* benchmark for ItorL, which is called the demo-navigation (DN) benchmark. In DN, we control a point agent from a start position to a target position in a maze, based on some expert demonstrations that can reach the target positions. The maze and target position can be changed between episodes. The agent can observe its $l$-step-length local views, while its current coordinate is optionally provided. In our experiment, the local view is calculated using 8 rays, each within 5 step length. This agent *does not capture the global map information*. Without utilizing the demonstrations, *it is impossible, under the given state space,* to find routes to the target positions for all maps. Besides, for each episode, the map will randomly generate some rectangular obstacles on the way to the target. These obstacles might not exist when the expert generates the demonstrations. *Thus the agent cannot exploit the demonstration*, i.e., repeat the actions in the demonstration without considering the current situation, to reach the target. We give an example of DN in Fig. 3. In the visualization, the start position is represented by a blue point, the target position by a green point, and the current agent position by a red point with red dashed lines representing the local views. Walls are indicated by black lines and obstacles by brown rectangles, which are not accessible to the agent. The gray points correspond to states in an expert demonstration. The details are in App. E.

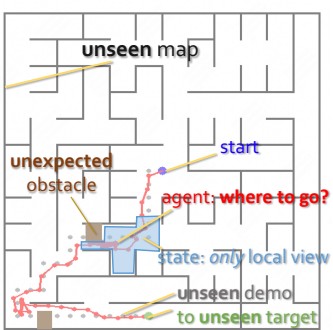

Figure 3: Illustration of the demo-navigation benchmark, where the red line is run by DAAC.

### 5.2 EXPERIMENT SETUP

**Tasks** Our primary focus is whether the policies exhibit out-of-the-box imitation capabilities beyond the demonstrations observed during the training. In our study, we create eight tasks within DN by varying three factors: (1) single-map versus multi-map navigation; (2) the presence or absence of obstacles; and (3) whether agent coordinates are provided. For each task, we gather demonstrations targeting different points. To validate the generalization capabilities, we withhold a portion of *new*

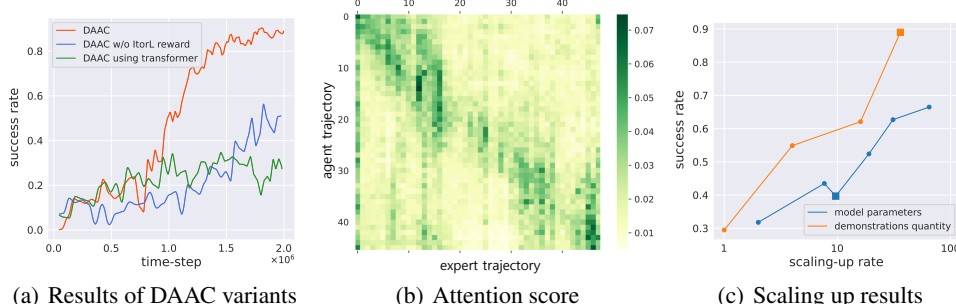

| (a) Results of DAAC variants | (b) Attention score | (c) Scaling up results |
| --- | --- | --- |

Figure 4: (a) Learning curves of DAAC variants; (b) The attention score map. The vertical axis represents the agent's trajectory, and the horizontal axis represents the expert's trajectory. The deeper the color in a row, the more attention the agent pays to the corresponding expert state. (c) The asymptotic performance of DAAC under different demonstration quantities and model parameters, where each unit in the x-axis denotes 60 demonstrations and 0.6 million parameters respectively. Please note that the x-axis is on a logarithmic scale. The *square markers* in the figure represent the performance of the default DAAC parameters we adopted.

*demonstrations* in each map for testing. Moreover, in the multi-map settings, we separately create *new maps* to collect demonstrations and evaluate the trained policies. More details are in App. E.

**Baselines** We compare DAAC with three main context-based learning approaches which also take demonstrations as inputs: (1) **DCRL** (Dance et al., 2021) embeds demonstrations with Transformer and trains policies with task-specific rewards for further improving the expert behavior via RL; (2) **TRANS-BC** (Dasari & Gupta, 2021) uses Transformer to extract representations from demonstrations and adopts BC for policy reconstruction. The auxiliary tasks for TRANS-BC like inverse dynamics loss on randomized image observation are removed since the state space in our tasks is low-dimensional with clear implications. (3) **CbMRL** (OpenAI et al., 2019; Peng et al., 2018) trains policies only with environment rewards. The demonstrations are simply embedded with a multi-layer GRU (Cho et al., 2014), which is the standard implementation of the framework in Alg. 1. All methods are trained for the same duration with the same parameter quantity to ensure fairness.

### 5.3 OUT-OF-THE-BOX IMITATION ABILITY IN UNSEEN SITUATIONS

We summarize all experimental results in Tab. 1. It's evident that DAAC dominates all tasks *with a large margin*, demonstrating its superior out-of-the-box imitation ability compared to existing baselines across all tasks. In the absence of coordinates, especially in multi-map scenarios, the performance of DAAC is not particularly ideal (considering a generalization success rate below 60% as the standard). This aligns with our expectations that, without coordinates, local views in a single trajectory cannot provide enough information for imitation, i.e., Prop A.2 is violated: In this case, any map may contain an arbitrary number of states with the same local views but different actual positions, making it difficult for the policy to distinguish them and make the correct decisions. This resembles a partially observable MDP, and we leave further investigation as future work.

On the other hand, we can see that both DCRL and CbMRL methods demonstrate a certain degree of imitation ability, which also confirms our claims in Sec. 4.1 that the context-based meta-RL framework can, in principle, handle ItorL. However, standard context-based policy architectures cannot fully utilize the demo information and are therefore not efficient enough. Although the Transformer-based DCRL overall performs better than the RNN-based CbMRL, both of them are less effective than our DA structure which is designed for ItorL scenarios. Finally, we find that the worst-performing method among all is TRANS-BC. Although this method also employs a Transformer, it fails to achieve satisfactory generalization in any task. This is because the demonstrations provided in our tasks are extremely limited. Solely relying on the BC framework without incorporating RL for environment interactions like other approaches makes it challenging to guarantee appropriate action outputs in unseen states.

### 5.4 EFFECTS OF THE DA ARCHITECTURE AND THE REWARD FUNCTION

We conduct ablation studies about the DA architecture and our ItorL reward on multi-map imitation tasks without obstacles and with coordinates provided. We construct two variants of DAAC: (1) DAAC using Transformer, where the actor and critic in DAAC are replaced with standard Transformer; (2) DAAC w/o ItorL reward, where DAAC just learns with the ending reward $R_\omega$. We test the trained policies directly on new maps and provide the learning curves in Fig. 4(a), we can observe that removing the imitator reward and replacing DA with Transformer results in a significant

Table 2: Success rate comparisons. The robot needs to imitate seen demonstrations and new demonstrations. The multi-task setting collects demonstrations equally from each manipulation task. We **bold** the best scores for each task.

| Domain | Complex Manipulation | | | | | | | | Complex Control Space | | | |
|---|---|---|---|---|---|---|---|---|---|---|---|---|
| Tasks | Grasping | | Stacking | | Collecting | | Multi-Task | | Reacher | | Pusher | |
| Demonstrations | seen | new_demo | seen | new_demo | seen | new_demo | seen | new_demo | seen | new_demo | seen | new_demo |
| DAAC | **0.98** | **0.84** | **0.77** | **0.84** | **0.99** | **0.61** | **0.89** | **0.45** | **0.98** | **0.95** | **0.96** | **0.94** |
| DCRL | 0.30 | 0.70 | 0.00 | 0.00 | 0.00 | 0.00 | 0.05 | 0.02 | 0.65 | 0.50 | 0.89 | 0.87 |
| TRANS-BC | 0.28 | 0.20 | 0.00 | 0.02 | 0.17 | 0.06 | 0.10 | 0.02 | 0.63 | 0.59 | 0.20 | 0.08 |
| CbMRL | 0.71 | 0.49 | 0.00 | 0.00 | 0.00 | 0.00 | 0.04 | 0.00 | 0.90 | 0.87 | 0.91 | 0.85 |

reduction in learning efficiency. Similar ablation results on robot manipulation tasks can be found in App. G. We also give detailed ablation studies ablation about the reward function, which is in App. G. The performance of DAAC using Transformer declines, indicating that without our DA architecture, the agent cannot fully utilize the demonstration information.

To further verify that DA stimulates the agent making decisions based on the discrepancy between the current state and the states in demonstrations, we visualize attention scores during the decision-making process in Fig. 4(b), which are products of the vectors of keys in demonstrations and the query of current states. Since the agent trajectory is similar to the expert trajectory, higher attention values mainly concentrate on the diagonal demonstrating that the agent actively matches expert states based on the matched state for decision making. More visualizations are provided in App. I.

## 5.5 THE POTENTIAL FOR FURTHER PERFORMANCE IMPROVEMENT WHEN SCALING UP

Inspired by the recent advances in large language models (OpenAI, 2023; Wei et al., 2022; Zhou et al., 2023), we investigate the potential for out-of-the-box imitation ability improvement when scaling up. In particular, we train DAAC policies with varying quantities of demonstrations and model parameters in multi-map imitation tasks involving obstacles. We test demonstration quantities in the coordinates-provided setting and model parameters in the no-coordinate setting and then verify the policies on new maps. We visualize experimental results in Fig. 4(c) and observe a log-linear increment of our model's performance with an increase in either data volume or model parameters. Particularly in the non-coordinate setting, increasing the model parameters leads to an around $2\times$ *improvement* in performance compared to the results shown in Tab. 1. These results provide strong evidence of the potential for performance improvement when scaling up the DAAC, and we plan to investigate further in future work.

## 5.6 APPLY DAAC TO COMPLEX TASKS

We deploy our DAAC method on robot tasks, including **Complex Manipulation**: The robot needs to imitate types of robotics tasks like object grasping, object stacking, object collecting, and mixed tasks in *clutter* environments, and **Complex Control Sapce:** We test the methods in the Reacher and Pusher environments (Towers et al., 2023). These environments feature variables diverse, including location, velocity, angular velocity, and so on, which exhibit substantial differences in magnitudes across dimensions. The details of the environments are in App. E.

We compare DAAC with its baselines and summarize the results in Tab. 2. Our method outperforms all baselines both on seen and new demonstrations, demonstrating that it is competent on more complex tasks. Note that, our method is the only one that can imitate all types of manipulation demonstrations and achieve satisfactory performance. Our method outperforms the baselines with high task completion rates, demonstrating its robustness in complex observation spaces.

# 6 DISCUSSION AND FUTURE WORK

We proposed a new topic, imitator learning (ItorL), which derives an imitator module to reconstruct task-specific policies out-of-the-box based on single expert demonstrations. We formulate the problem and propose a practical solution, *Demo-Attention Actor-Critic* (DAAC). We apply DAAC to both demo-navigation tasks and complex robot manipulation tasks, which shows that DAAC outperforms previous IL methods with large margins both on training and unseen-tasks testing.

We believe that ItorL is a novel and challenging topic for the IL community, and there might be many interesting ItorL applications in autonomous vehicles and robotics. The scaling-up experiments in Sec. 5.5 also demonstrate the potential of DAAC in solving larger-scale problems, which we will investigate in our future work. Currently, the limitations of DAAC include: (1) in without-coordinates scenarios, which imply a "POMDP" problem, DAAC is not particularly ideal; (2) the inference's compute resource requirement intrinsically increases as the number of demonstrations grows because of the self-attention mechanism; and (3) the imitator ability in far-away states.

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

# Appendix

## Table of Contents

## A    DEMONSTRATION QUANTITY REQUIREMENTS FOR IMITATOR LEARNING

Without further assumptions on task space $\Omega$, it is always easy to construct ill-posed problems that it is impossible for a unified imitator policy $\Pi(a|s, \mathcal{T}_\omega)$ to reconstruct all of the expert policies unless the expert demonstration set $\mathcal{T}$ does cover the whole state-action space. However, in many applications, it is unnecessary for $\Pi$ to imitate policies for any $\mathcal{M}$. Here we give one practical task set that enables imitator learning (ItorL) through only one demonstration.

**Definition A.1** ($\tau_\Omega$-tracebackable MDP set). For an MDP set $\mathbb{M} := \{\mathcal{M}_\omega \mid \omega \in \Omega\}$, if there exists a unified goal-conditioned policy $\beta(a|s, g)$, $\forall \mathcal{M}_\omega \in \mathbb{M}$, for any $\tau_\omega$, we have $\forall s_i \in \tau_\omega$ or $\forall s_0 \in R(d_0)$, $\exists g_j \in \tau_\omega$, $\beta(a|s, g_j)$ can reach $g_j$ from $s = s_i$ within finite timesteps, where $R(X)$ is the state set in $X$, $i$ and $j$ denote the timestep of states in $\tau$ and $j > i$, then $\mathbb{M}$ is a $\tau_\Omega$-tracebackable MDP set.

$\tau_\Omega$-tracebackable MDP set depicts the similarity of the tasks in $\mathbb{M}$ through the demand of the policy $\beta(a|s, g)$. It means that although the transition process and the initial distribution are stochastic and different among $\mathbb{M}$, there exists a goal-conditioned policy $\beta$ that for any $\mathcal{M}_\omega$, we can guide the agent turn back to some states in the demonstrations. For example, different navigation tasks will have similar decisions in similar traffic conditions even in different terrains. Thus even if the vehicle has to veer off the demonstrations for handling some unexpected situations, it usually can turn back after some timesteps.

Based on the definition, we give an $\mathbb{M}$ formulation that can find a unified imitator policy $\Pi$ from one demonstration.

**Proposition A.2** (1-demo imitator availability). *If $\mathbb{M} := \{\mathcal{M}_\omega \mid \omega \in \Omega\}$ is a $\tau_\Omega$-tracebackable MDP set, there exists at least a unified imitator policy $\Pi(a|s, \mathcal{T}_\omega)$ that can accomplish any task in $\mathbb{M}$ only given one corresponding demonstration, i.e., $|\mathcal{T}_\omega| = 1$.*

*Proof.* Since $R_\omega$ in $\mathcal{M}_\omega$ is an ending reward function of trajectories, given any expert demonstration $\tau_\omega$, we know:

$$R_\omega(s, a) = \begin{cases} c, & s = s_t \\ 0, & (s, a) \in \tau_\omega \text{ and } s \neq s_t \\ \text{unkown} & \text{otherwise} \end{cases}$$

that is, any policy can accomplish the task in $\mathcal{M}_\omega$ if it can reach the last state $s_t$ of $\tau_\omega$, where $c$ is the reward for accomplishing the task.

Since $\mathbb{M} := \{\mathcal{M}_\omega \mid \omega \in \Omega\}$ is a $\tau_\Omega$-tracebackable MDP set, there exists a unified goal-conditioned policy $\beta(a|s, g)$, $\forall \mathcal{M}_\omega \in \mathbb{M}$, for any $\tau_\omega$ which can accomplish the task in $\mathcal{M}_\omega$, we have $\forall s_i \in \tau_\omega$ or $\forall s_0 \in R(d_0)$, $\exists g_j \in \tau_\omega$, $\beta(a|s, g_j)$ can reach $g_j$ from $s = s_i$ within finite timesteps, where $R(X)$ is the state set in $X$, $i$ and $j$ denote the timestep of states in $\tau$ and $j > i$. We can construct a unified imitator policy by (1) searching a $g_j \in \tau_\omega$ that can be reached by $\beta(a|s, g_j)$ from current state $s_i$ within finite timesteps, where $j > i$; (2) executing $\beta(a|s, g_j)$ until reaching $g_j$; (3) repeat (1) and (2) to the end. When deployed, for any $\mathcal{M}_\omega$, in the beginning, $s_0 \sim d_0$, thus the agent will reach one of the state $s_i \in \tau_\omega$ after finite timesteps, where $i > 0$. Since $s_i \in \tau_\omega$, following $\beta(a|s, g_j)$, the agent will arrive another state $s_j \in \tau_\omega$. The process will be repeated until the agent reaches the last state $s_t$. Once $R_\omega(s_t, \cdot) = c$, the task is accomplished.    $\square$

Although we focus on 1-demo imitator availability, note that the 1-demo imitator availability can be extended to the "$n$-demo" case by extending $\tau_\Omega$-tracebackable MDP set to "$\mathcal{T}_\Omega$-tracebackable MDP set".

Fig. 5 gives a vehicle navigation illustration for the proposition, where all tasks in $\mathbb{M}$ ask the vehicle to reach some locations based on its coordinates and local views. We first consider a simple case in which the initial state is deterministic and is the same as the first state in $\tau_\omega$. In this case, even if a truck might be parked unexpectedly (states unvisited in the demonstrations), relying on the local-view information, for any $\tau_\omega$, we have a unified goal-conditioned policy $\beta(a|s, g)$, i.e., closing to some of the successor expert states without collision, that can drive the vehicle to be close to the locations in $\tau_\omega$. With policy $\beta$, there exists at least a unified imitator policy $\Pi(a|s, \mathcal{T}_\omega)$ that can accomplish any task in $\mathbb{M}$ only given one corresponding demonstration: repeatedly traces a reachable successor state $g_j \in \tau_\omega$ and uses $\beta$ to guide the agent until reaching the ending state.

In the following, we consider a counter-example where the agent state can be put to untrackbackable states, e.g., the square point in Fig. 5, for some unforeseen reasons. In this case, if the local view is limited and cannot reach the location of entrances and the entrance might exist *either* in A or B, it is impossible to construct a unified goal-conditioned policy $\beta(a|s,g)$ since in the square point, the correct way to trace back to the demonstrations is agnostic (can be in left or right).

Note that the trace-backable property relies on the information we have from the states, e.g., with global map information in the state space, the unified goal-conditioned policy can be constructed by planning a trajectory in the map then the above task set is trace-backable.

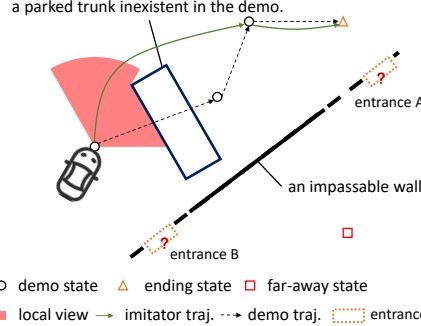

Figure 5: A vehicle navigation example. "traj." is the abbreviation of "trajectory".

# B    ITORL REWARD FROM DEMONSTRATIONS

A theoretical analysis in Ciosek (2022) shows that, for deterministic experts, IL can be done by RL with a constructed stationary reward: $R_{\text{int}}(s,a) = \mathbb{I}[(s,a) \in \mathcal{T}]$, where $\mathbb{I}[\cdot]$ denotes the indicator function and $\mathcal{T}$ is the expert demonstration. In practice, the constructed reward function:

$$R_{\text{IL}}(s,a) = 1 - \min_{(s',a') \in \mathcal{T}} d_{\ell_2}((s,a),(s',a'))^2, \tag{2}$$

which is a practical imitation reward $R_{\text{int}}$ that can also imitate the experts in several benchmark tasks. Here $d_{\ell_2}(\cdot,\cdot)$ denotes the $\ell_2$ distance of two normalized vectors.

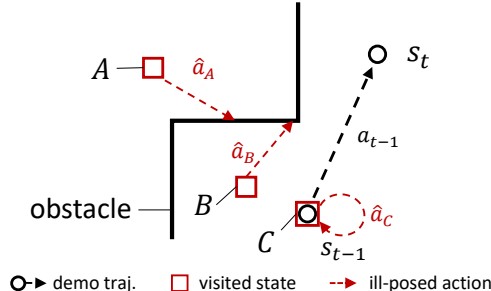

Figure 6: Illustration of the ill-posedness of $R_{\text{IL}}$. $A$, $B$, and $C$ denote states, and red dashed arrows ($\hat{a}_A$, $\hat{a}_B$, and $\hat{a}_C$) denote the corresponding ill-posed sub-optimal actions to earn more the cumulative $R_{\text{IL}}$ rewards. The agent fails on hitting the obstacle. $s_t$ is the last state, also the target state for task completion.

Inspired by this, we propose to construct a stationary imitator reward $R_{\text{Itor}}$ to embed IL into the RL process, i.e., replacing the reward function $R_\omega$ in Alg. 1 (Line 7) with $R_{\text{Itor}}$. First, we observe that $R_{\text{IL}}$ and $R_{\text{int}}$ can reconstruct the expert policy only when we have a diverse enough dataset $\mathcal{T}$ which covers the state-action space. When only with limited demonstrations, the reward function will be ill-posed in three aspects. We depict that based on the illustration in Fig. 6: (1) $A$ state: if the minimum-distance tuple in Eq. 2 is far away from the visited state, e.g., $(s_{t-1}, a_{t-1})$ in Fig. 6, the action that reduces the $\ell_2$-norm between the next state and $s_{t-1}$ might ignore the impassable terrains between states and finally hit the obstacle; (2) $B$ state: even if the action to reduce the $\ell_2$-norm between the next state of $B$ and $s_{t-1}$ is correct to go back to the demonstration, to reduce the $\ell_2$-norm between actions $a_{t-1}$ and the current action at the same time, the derived action might be biased by $a_{t-1}$ and finally lead to an unsafe state; (3) $C$ state: even if the state perfectly matches the one in the demonstration, the agent still has the potential to stay where it is until it reaches the maximum episode length, as $R_{\text{IL}}$ might be greater than 0. To handle the above problems, we

construct a new imitator reward function $R_{\text{Itor}}(s, a)$ via:

$$R_{\text{Itor}}(s, a) := 1 - \min \left\{ d(\bar{s}, s)^2 + \frac{d(\bar{a}, a)^2}{\exp(d(\bar{s}, s)^2)}, \eta \right\} + \alpha R_\omega(s, a), \qquad (3)$$

where $(\bar{s}, \bar{a}) = \arg\min_{(s', a') \in \mathcal{T}} d(s, s')^2$, $\eta$ is a hyperparameter that clips the distance penalty calculated based on the too-far state pairs into a fixed constant, $\alpha = 1/(c(1 - \gamma))$ is a rescale coefficient, and $c$ is the reward for accomplishing the task defined in $R_\omega$.

$R_{\text{Itor}}(s, a)$ uses a clipping term $\eta$ to make the imitation rewards based on the too-far state pairs invalidated to avoid the potential misleading (to handle the "$A$-state" case). A reweighting item $1/\exp(d(\bar{s}, s)^2)$ is used for the action's distance computation to adaptively adjust the weight of rewards on action matching. This is a heuristic reweighting term to avoid the agent overly penalizing for not strictly following the expert action when its current state is far from the demonstration states and chooses to turn back (to handle the "$B$-state" case). The necessity of the reweighting term stems from its pivotal role in preventing undesired behaviors in situations where the agent strictly adheres to the demonstrated actions due to state bias. By incorporating the reweighting term, we ensure that the agent does not blindly follow the demonstrations, thereby reducing the risk of unintended consequences. $\alpha$ rescales the ending rewards, which makes the discount on delay to get the ending reward larger than the bonus of repeatedly collecting the immediate rewards, i.e., $\alpha c > 1 - \epsilon + \gamma \alpha c$, where $\epsilon$ denotes a larger-than-zero penalty contributed by the second item in Eq. 3 (to handle the "$C$-state" case).

Note that although we give several tricks to make $R_{\text{Itor}}$ give reasonable rewards in the state-action space, it is still inevitable to output ill-posed rewards in some corner cases. Hence, the ending reward is essential, as it helps the agent focus more on task completion rather than repeatedly collecting $R_{\text{IL}}$ rewards. The large coefficient $\alpha$ on the task-specific reward $R_\omega$ makes the policies always focus on completing the tasks rather than repeatedly collecting $R_{\text{IL}}$ rewards. In this situation, $R_{\text{Itor}}$ just serves as a crucial signal by providing a dense reward, enabling the agent to closely follow the demonstrations and accomplish tasks effectively during the early stages. We leave a theoretical-grounded reward function design as future work.

## C  RELATED WORK

### C.1  IMITATION LEARNING

Imitation learning (IL) focuses on training a policy with action labels from expert demonstrations. There are two mainstream approaches for IL, namely behavior cloning (BC) (Pomerleau, 1991; Ross & Bagnell, 2010) and inverse reinforcement learning (IRL) (Ng & Russell, 2000). The former BC converts IL into a supervised paradigm by minimizing the action probability discrepancy with Kullback Leibler (KL) divergence between the actions of the imitating policy and the demonstration actions. The latter IRL fashion learns the hidden reward function behind the expert policy to avoid the impact of compounding errors.

Since IL can learn directly from already collected data, it is widely adopted by complex domains like game playing (Ross & Bagnell, 2010), autonomous driving (Chen et al., 2019; Pan et al., 2018), and robot manipulation (Xie et al., 2020). Although achieving impressive performances, we observe that in many applications, what humans require is the ability to perform many different tasks out of the box, through very limited demonstrations of corresponding tasks, instead of imitating from scratch based on a mass of demonstrations. Adapting the trained policy to unseen tasks is beyond the capability of pure IL, which is designed for single-task learning.

### C.2  META-IMITATION LEARNING

Meta-IL includes few-shot meta-IL and one-shot meta-IL. Few-shot meta-IL aims to get a generalizable policy that can complete new tasks with only a few expert trajectories. The mainstream solutions utilize model-agnostic meta-learning (MAML) (Finn et al., 2017a) to learn initial task parameters and fine-tune them via a few steps of gradient descent to satisfy new task needs (Finn et al., 2017b; Li et al., 2021; Yu et al., 2018). However, these approaches need extra computation infrastructure for gradient update and determining a suitable amount of fine-tuning steps before deployment (Finn et al., 2017a). One-shot meta-IL achieves generalizable imitation through

context-based policy models (Dasari & Gupta, 2021; Duan et al., 2017; Mandi et al., 2022), such as Transformer (Vaswani et al., 2017), that take demonstrations as input. The core idea is to extract representations of demonstrations through these powerful fitting abilities of neural networks, then use BC to reconstruct the imitation policy. However, the demonstrations for imitation are limited, the inevitable prediction errors on unseen states and the compounding errors of BC (Ross et al., 2011) hurt the capacities of these methods, especially in generalizing to new tasks (Mandi et al., 2022). Different from one-shot IL, In ItorL setting, we argue that the interactions with simulators are allowed, so that we have more potential ways to handle the generalization ability with less demonstration. Our ItorL method also utilizes a context-based model to achieve the out-of-the-box imitation ability. Instead of using BC, we integrate IL into the RL process, which allows the agent to interact with the environment. This approach can regularize the policy behavior when facing states unvisited in demonstrations.

The main differences between the Meta-IL approach and our approach are primarily in two aspects: (1) **no need for fine-tuning**: Our objective is to create an imitator policy, $\Pi(a|s, \mathcal{T}_\omega)$, informed solely by a pre-collected expert demonstration set, without requiring any fine-tuning. During deployment, this policy simply takes in the relevant demonstration $\tau_\omega$ to generate the appropriate action for any given state. In contrast, most few-shot IL techniques, like MAML (Finn et al., 2017a), necessitate fine-tuning for the target task. (2) **imitation with single demonstration**: Our deployment only requires a single trajectory for imitation. While some algorithms might achieve imitation without fine-tuning using transformer architectures, both MAML and Transformer-BC (Dasari & Gupta, 2021) necessitate a substantial volume of trajectories for target adaptation during deployment. Our ability to achieve imitation with even fewer trajectories comes from our interaction with simulators, enabling the implicit learning of cross-task general imitation behavior.

### C.3 Combination of Imitation Learning and Reinforcement Learning

We are not the first study to combine IL and RL. Previous studies have combined these for different proposes: Hester et al. (2018) leverage small sets of demonstrations for deep q-learning which massively accelerates the learning process. Rajeswaran et al. (2018) use demonstrations to reduce the sample complexity of learning dexterous manipulation policy and enable natural and robust robot movement. Fujimoto & Gu (2021) add BC to the online RL algorithm TD3 (Fujimoto et al., 2018) for advanced offline RL performance. Our method extends the ideas of combining IL and RL to handle a new problem: a multi-policy imitation problem based on limited demonstrations.

### C.4 Context-based Meta Reinforcement Learning

Besides IL, context-based policy models are also widely used in meta-RL. Building a representative context enables a single agent of learning meta-skills and identifying new tasks. Goal-conditioned RL (Florensa et al., 2018; Nair et al., 2020) is the most direct way to build a context-based meta-policy, which scales a single agent to a diverse set of tasks by informing the agent of the explicit goal contexts, e.g., the target to go or the object to pick. The demonstrations can be regarded as an informative "goal" for IL tasks. The demonstration sequence not only tells the agent which task to accomplish but also the way to accomplish it.

Some other works collect interaction trajectories from the environment for understanding the task identity. Chen et al. (2021); Luo et al. (2022); Nagabandi et al. (2019); OpenAI et al. (2019); Peng et al. (2018) use a end-to-end architecture for environment-parameter representation and adaptable policy learning. A recurrent neural network is introduced for environment-parameter representation, then the context-aware policy takes actions based on the outputs of RNN and the current states. Rakelly et al. (2019) share the same end-to-end architecture and design a new neural network to represent the probabilistic latent contexts of the environment parameters. Instead of collecting trajectories from the environment for identifying the task, we mine the information from the static expert trajectories to identify the expert policy which can accomplish the task.

Demonstration-conditioned RL (DCRL) (Dance et al., 2021) takes sub-optimal demonstrations as input and seeks to further improve demonstration behavior via RL. Yeh et al. (2022) adopt a similar idea to solve unseen compound robot tasks that contain multiple stages by retrieving from demonstrations. Instead of taking demonstration as the base for policy improvement, ItorL aims to fully utilize the demonstrations to imitate the expert policy for each task.

# D IMPLEMENTATION DETAILS

## D.1 ACHITECTURE DETAILS

We give the demonstration-based attention architecture for the critic in Fig. 7, and related hyper-parameters of the architecture in Tab. 3.

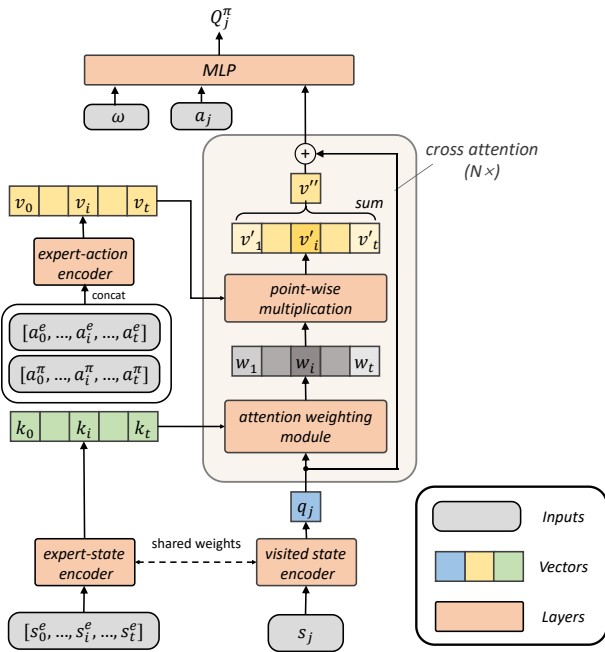

Figure 7: The architecture of demonstration-based attention for the critic. $[s_0^e, ..., s_i^e, ..., s_t^e]$ and $[a_0^e, ..., a_i^e, ..., a_t^e]$ denote state and action list in an expert demonstration. $a_0^\pi, ..., a_i^\pi, ..., a_t^\pi$ denote the output action of current actor $\Pi$ on $s_0^e, ..., s_i^e, ..., s_t^e$ respectively. $a_j$ is the output action of $\Pi$ on $s_j$. $s_j$ is the visited state of the actor at timestep $j$. Inspired by OpenAI et al. (2019); Miki et al. (2022), we feed the task parameter $\omega$ to the critic for $Q$ value prediction. It is valid because the critic will not be used when deployed, and $\omega$ gives important information for value inference. We use $q, k, v$ to denote the query, key, and value vectors of an attention module. $N \times$ denote a $N$-layer cross-atttention module, which take the output $v''$ of the last layer as the input $q_j$ of the next layer.

where $(\bar{s}, \bar{a}) = \arg \min_{(s', a') \in \mathcal{T}} d_{\ell_2}(s, s')^2$, $\eta$ is a hyperparameter that clips the distance penalty calculated based on the too-far state pairs into a fixed constant, $\alpha = 1/(c(1 - \gamma))$ is a rescale coefficient, and $c$ is the reward for accomplishing the task defined in $R_\omega$.

In DA architecture, we introduce three encoders for expert actions, expert states, and visited states respectively, where the encoders of expert states and visited states share the same weights. The detailed architecture is shown in Fig. 8.

Table 3: DAAC Hyper-parameters.

| Parameter | Value |
|---|---|
| learning rate ($\lambda$) | $5 \cdot 10^{-5}$ |
| discount ($\gamma$) | 0.99 |
| replay buffer size | $10^5$ |
| number of hidden units per layer | 256 |
| number of samples per minibatch | 256 |
| optimizer | RMSprop |
| *Actor* | |
| encoder layer number ($K$) | 3 |
| cross-attention layer number ($N$) | 6 |
| embedding dimension | 128 |
| *Critic* | |
| encoder layer number ($K$) | 4 |
| cross-attention layer number ($N$) | 4 |
| embedding dimension | 128 |
| *ItorL Rewards for Demo-Navigation* | |
| rescale coefficient ($\alpha$) | 100 |
| penalty threshold ($\eta$) | 2 |
| *ItorL Rewards for Robot Manipulation* | |
| rescale coefficient ($\alpha$) | 200 |
| penalty threshold ($\eta$) | 2 |

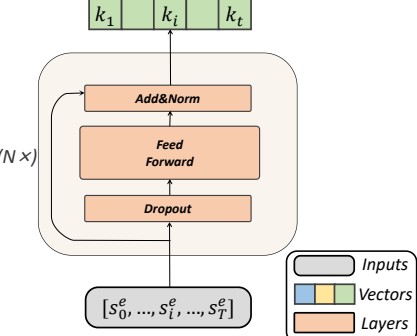

Figure 8: The encoder architecture employed in the DA model. As an example, we consider the input sequence $[s_0^e, ..., s_i^e, ..., s_t^e]$. However, it is worth noting that this architecture can also accommodate $[a_0^e, ..., a_i^e, ..., a_t^e]$ and $[a_0^\pi, ..., a_i^\pi, ..., a_t^\pi]$ as inputs. The encoder leverages the Transformer backbone, which incorporates three layers: dropout, feedforward, and add&norm. These layers are organized using the residual connection mechanism. The input sequence passes through $N$ stacked blocks, converting it into key vectors.

## D.2 TRAINING DETAILS

---

**Algorithm 2** Demo-Attention Actor-Critic for ItorL

---

**Input:** A task set $\mathbb{M}_{\text{train}}$, and a demonstration set $\{\mathcal{T}_{\omega_i}\}$ for each task $\mathcal{M}_{\omega_i} \in \mathbb{M}_{\text{train}}$
**Process:**
1: Initialize Actor $\pi_\phi$, Critic $Q_\theta$ and a replay buffer $\mathcal{B}$
2: **for** $1, 2, 3, ...$ **do**
3:     Sample a task $\mathcal{M}_\omega$ from the sampling strategy $P(\mathbb{M}_{\text{train}})$
4:     Get the *single* expert trajectory $\tau_\omega$ since $|\mathcal{T}_\omega| = 1$
5:     **for** $j = 1, 2, 3, ..., H$ **do**
6:         Sample an action $a_j \sim \pi_\phi(a|s_j; \tau_\omega)$
7:         Rollout one step $s_{j+1} \sim \mathcal{M}_\omega(s|s_j, a_j)$, get the reward $r_j = R_{\text{Itor}}(s_j, a_j)$
8:         Add $(s_j, a_j, r_j, s_{j+1}, \tau_\omega)$ to $\mathcal{B}$
9:     **end for**
10:     **for** each update step **do**
11:         update Critic $\theta \leftarrow \theta - \lambda \nabla J_Q(\theta)$
12:         update Actor $\phi \leftarrow \phi - \lambda \nabla J_\pi(\phi)$
13:     **end for**
14: **end for**

---

We use the SAC (Haarnoja et al., 2018) algorithm to update the DA-actor and DA-critic. The goal of SAC also maximizes the expected entropy return beyond the objective of a standard RL agent which maximizes the expected sum of rewards:

$$J(\pi) = \sum_{t=0}^{T} \mathbb{E}_{(s_t, a_t) \sim \rho_\pi}[r(s_t, a_t) + \alpha \mathcal{H}(\pi(\cdot|s_t))], \tag{4}$$

where $\mathcal{H}(\pi(\cdot|s_t))$ is the entropy value of the policy distribution. For learning the maximum entropy, a policy alternates between policy evaluation and policy improvement. For policy evaluation of a fixed policy, we can obtain its soft state value function by iteratively applying the Bellman update:

$$V(s_t) = \mathbb{E}_{a_t \sim \pi}[Q(s_t, a_t; \tau_\omega) - \alpha log\pi(a_t|s_t; \tau_\omega)]. \tag{5}$$

And we can execute critic update through collected buffer data and the objective:

$$J_Q(\theta) = \mathbb{E}_{(s_t, a_t, \tau_\omega) \sim \mathcal{D}} \left[ \frac{1}{2} \left( Q_\theta(s_t, a_t; \tau_\omega) - \hat{Q}(s_t, a_t) \right)^2 \right], \tag{6}$$

with

$$\hat{Q}(s_t, a_t) = r_t + \gamma \mathbb{E}_{s_{t+1} \sim p}[V(s_{t+1})], \tag{7}$$

And we can execute policy improvement through collected buffer data and the objective:

$$J_\pi(\phi) = \mathbb{E}_{s_t \sim \mathcal{D}} D_{\text{KL}} \left( \pi_\phi(\cdot|s_t; \tau_\omega) \,\middle\|\, \frac{\exp(Q_\theta(s_t, \cdot; \tau_\omega))}{Z_\theta(s_t)} \right), \tag{8}$$

where the partition function $Z_\theta(s_t)$ normalizes the distribution. We adopt one policy (actor) network, two Q-networks (critic), and two target Q-networks for SAC training. Each network consists of one demonstration-based attention module for task-information extraction and projects the task embedding into actions.

For learning robust policies, we randomly choose a state from the given demonstration as a start and add a disturbance of $0.1 \times N(0, 1)$ to this state coordinate. We maintain a separate buffer for each demonstration and gather a batch of training data from 5 different buffers. To accelerate the training process, we also add demonstration data which takes $20\%$ of the batch size for joint training. For fair comparisons, all the baselines we compared followed the above setting. The detailed hyperparameters used for our ItorL method training are summarized in Tab. 3.

# E ENVIRONMENT DESCRIPTION

## E.1 DEMO-NAVIGATION ENVIRONMENT

We include details of our two-dimensional maze environment for navigation tasks, where the maze layout takes a size of 24×24. The maze is generated by randomly traversing all the cells in a Depth-First manner with path width 2. The path in the maze is connected, thereby our environment is $\tau_\Omega$-tracebackable and 1-demo imitator available, which satisfies our ItorL needs. We fixed the starting point as the center of the map. The expert trajectories can also be obtained by a Depth-First search. We can formulate this environment as Markov Decision Process, which can be presented as a tuple $(\mathcal{S}, \mathcal{A}, T, R)$.

**State space** $\mathcal{S}$: The maze state consists of the $(x, y)$ coordinate and the local view of the agent along 8 different directions with an equal interval $\pi/4$. For the simplest task where coordinates are provided and no obstacles exist, the local view length $l$ is set to 1.5; otherwise 5 for observing the surrounding environment changes.

**Action space** $\mathcal{A}$: The agent is able to take action $(\Delta_x, \Delta_y)$ which are continuous values in the range of $[-1, 1]$.

**Transition function** $T$: When applied with the action $(\Delta_x, \Delta_y)$ at the coordinate $(x, y)$, the agent translates itself to the $(x + \Delta_x, y + \Delta_y)$ coordinate. Some obstacles, which have lengths in the range of $[1.1, 1.3]$ and widths of 1.35, may appear in the demonstration path. The obstacle will be generated with a fixed probability $p = 0.1$ for each demonstration step and with a maximum number of 4. Once hits the wall or the obstacles, the agent will be dead and the trajectory will be terminated.

**Reward function** $R_\omega$: We only have a simple reward function $R_\omega$ which indicates the ending of trajectories, e.g., $c$ for reaching the target goal, 0 for failure in 50 timesteps, and $-c$ for dead, where $\omega$ is the goals we set.

Due to the unavailability of the global map, the agent is expected to follow the demonstration and reconstruct the expert's behavior to reach the goal, as illustrated in Fig. 9(a). Beyond this, some unexpected obstacles may occur, which results in that strictly following the demonstration no longer working, as shown in Fig. 9(b). The agent is expected to learn robust policies that can bypass obstacles and finish the task, based on imitating the given demonstration.

For single-map scenarios, we randomly choose 90% of all demonstrations (290 demos for each map) for training while the left is for evaluation. For multi-map imitation, we generate 240 different maps and only select a small number of 10 training demonstrations from each map. We treat the remaining *new demonstrations* to verify generalization. Besides, we also create 10 *new maps* separately to verify whether the agent trained on the multi-map scenarios works.

## E.2 DEMO-MANIPULATION ENVIRONMENT

We introduce details of our robot manipulation environment to verify the imitation ability of our method across different tasks. This environment consists of three types of robotics manipulations, namely object grasping, object stacking, and object collecting. We provide illustrations in Fig. 10, where the workspace is a 50 cm × 70 cm area. To collect demonstrations, we instruct the robot to execute predefined primitives in sequence. For instance, grasping a single object comprises three primitives: 1) moving the gripper to the object; 2) closing the gripper; 3) moving the gripper to the target. We present the Markov formalization of this environment in the following.

**State space** $\mathcal{S}$: The robot manipulation state includes the absolute position of the robot gripper, the absolute positions of the objects, and the relative positions of the gripper fingers.

**Action space** $\mathcal{A}$: The agent is able to take action $(\Delta_x, \Delta_y, \Delta_z, \Delta_c)$, each of which is continuous value in the range of $[-1, 1]$. The first three dimensions indicate the desired increment in the gripper position at the next timestep, while the last dimension controls the positions of the gripper fingers.

**Transition function** $T$: When applied with the action $(\Delta_x, \Delta_y, \Delta_z)$ at the coordinate $(x, y, z)$, the robot gripper moves to the new coordinate $(x + \Delta_x, y + \Delta_y, z + \Delta_z)$. If $\Delta_c > 0$, the gripper opens; otherwise, it closes. The task is considered failed if any object falls off the desk.

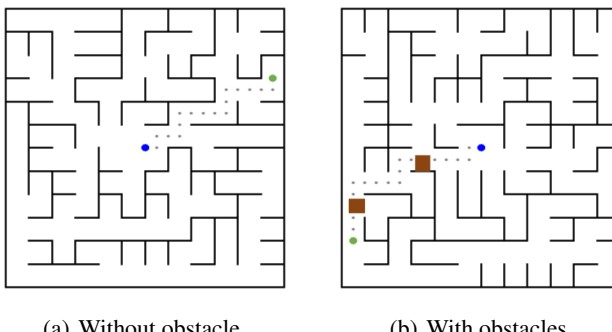

(a) Without obstacle.          (b) With obstacles.

Figure 9: Fig. 9(a) shows a demonstration sample on a map without obstacle, where we fix the start position as the center of the map (colored in blue), while the agent is expected to reach the specified goal colored in green. The agent should follow the expert trajectory to achieve the goal due to the unavailability of the global map. Fig. 9(b) shows a demonstration sample on a map with obstacles. Here strictly imitating the expert trajectory cannot well handle unexpected situations, e.g., the agent is blocked by obstacles when it follows the given demonstration. Beyond pure imitation, the agent should also explore the environment to learn robust policies.

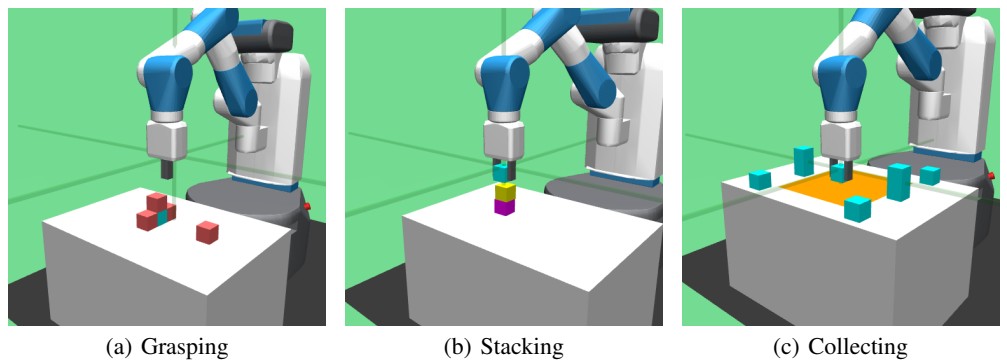

(a) Grasping          (b) Stacking          (c) Collecting

Figure 10: Various tasks of robot manipulation. (a): Grasp the blocked target object (cyan). (b): Stack the objects. (c): Collect the objects scattered over the desk together to the specified area (yellow).

**Reward function** $R$: Similar to the demo-navigation environment, our reward function $R$ is simple and only indicates the end of trajectories. That is, we use $c$ to indicate task accomplishment, $-c$ for failure, and 0 for all other situations. The criteria for accomplishing each task differs. In the object grasping task, the robot needs to grasp the target object without colliding with other objects. In the object stacking task, the robot must stack three blocks together, which are initially placed anywhere on the workspace. Lastly, in the object collecting task, the robot needs to collect all objects scattered over the desk and place them in a specified area.

We generate 300 demonstrations for each task, of which 60 are used for training and the remaining 240 are for testing. To verify the imitation ability of our method across multiple manipulation tasks, we generate 100 demonstrations for each of the three tasks. We randomly select 20 demonstrations from each task for training and leave 240 new demonstrations for the test.

### E.3   PUSHER AND REACHER ENVIRONMENT

For the pusher task, the state comprises the positions and velocities of the robot's joints (a total of 7), as well as the position of the robot tip arm and the manipulated object. The agent accomplishes the task by taking actions that modify the rotation of each joint and drive the robot to push the object

to the specified position. The specific contents of the state space and action space are provided in the following:

Table 4: Details of Pusher observation space.

| Num | Observation | Min | Max | Name | Joint | Unit |
|---|---|---|---|---|---|---|
| 0 | Rotation of the panning the shoulder | -Inf | Inf | r_shoulder_pan_joint | hinge | angle (rad) |
| 1 | Rotation of the shoulder lifting joint | -Inf | Inf | r_shoulder_lift_joint | hinge | angle (rad) |
| 2 | Rotation of the shoulder rolling joint | -Inf | Inf | r_upper_arm_roll_joint | hinge | angle (rad) |
| 3 | Rotation of hinge joint that flexed the elbow | -Inf | Inf | r_elbow_flex_joint | hinge | angle (rad) |
| 4 | Rotation of hinge that rolls the forearm | -Inf | Inf | r_forearm_roll_joint | hinge | angle (rad) |
| 5 | Rotation of flexing the wrist | -Inf | Inf | r_wrist_flex_joint | hinge | angle (rad) |
| 6 | Rotation of rolling the wrist | -Inf | Inf | r_wrist_roll_joint | hinge | angle (rad) |
| 7 | Rotational velocity of the panning the shoulder | -Inf | Inf | r_shoulder_pan_joint | hinge | angular velocity (rad/s) |
| 8 | Rotational velocity of the shoulder lifting joint | -Inf | Inf | r_shoulder_lift_joint | hinge | angular velocity (rad/s) |
| 9 | Rotational velocity of the shoulder rolling joint | -Inf | Inf | r_upper_arm_roll_joint | hinge | angular velocity (rad/s) |
| 10 | Rotational velocity of hinge joint that flexed elbow | -Inf | Inf | r_elbow_flex_joint | hinge | angular velocity (rad/s) |
| 11 | Rotational velocity of hinge that rolls the forearm | -Inf | Inf | r_forearm_roll_joint | hinge | angular velocity (rad/s) |
| 12 | Rotational velocity of flexing the wrist | -Inf | Inf | r_wrist_flex_joint | hinge | angular velocity (rad/s) |
| 13 | Rotational velocity of rolling the wrist | -Inf | Inf | r_wrist_roll_joint | hinge | angular velocity (rad/s) |
| 14 | x-coordinate of the fingertip of the pusher | -Inf | Inf | tips_arm | slide | position (m) |
| 15 | y-coordinate of the fingertip of the pusher | -Inf | Inf | tips_arm | slide | position (m) |
| 16 | z-coordinate of the fingertip of the pusher | -Inf | Inf | tips_arm | slide | position (m) |
| 17 | x-coordinate of the object to be moved | -Inf | Inf | object (obj_slidex) | slide | position (m) |
| 18 | y-coordinate of the object to be moved | -Inf | Inf | object (obj_slidey) | slide | position (m) |
| 19 | z-coordinate of the object to be moved | -Inf | Inf | object | cylinder | position (m) |

Table 5: Details of Reacher observation space.

| Num | Action | Control Min | Control Max | Name | Joint | Unit |
|---|---|---|---|---|---|---|
| 0 | Rotation of the panning the shoulder | -2 | 2 | r_shoulder_pan_joint | hinge | torque (N m) |
| 1 | Rotation of the shoulder lifting joint | -2 | 2 | r_shoulder_lift_joint | hinge | torque (N m) |
| 2 | Rotation of the shoulder rolling joint | -2 | 2 | r_upper_arm_roll_joint | hinge | torque (N m) |
| 3 | Rotation of hinge joint that flexed elbow | -2 | 2 | r_elbow_flex_joint | hinge | torque (N m) |
| 4 | Rotation of hinge that rolls the forearm | -2 | 2 | r_forearm_roll_joint | hinge | torque (N m) |
| 5 | Rotation of flexing the wrist | -2 | 2 | r_wrist_flex_joint | hinge | torque (N m) |
| 6 | Rotation of rolling the wrist | -2 | 2 | r_wrist_roll_joint | hinge | torque (N m) |

## F THE ROBUSTNESS ON THE FARAWAY STATES

In general ItorL scenarios, the task set we faced might violate the assumptions defined in Def. 2.1, for example, the transition function might, with some probabilities, lead the agent to some states that are "significantly far away from" the expert states. More specifically, in these faraway states, we cannot trace back to the demonstrations just by relying on the current states' information; e.g., in the Maze-navigation benchmark, if the current states' coordinations are far away from the demonstrations, the way back to the demonstrations depends on the walls' location in the maps, which is unseeable to the agent, thus the agent cannot find a unified behavior to back to the demonstration and reach the goal. In this section, we conducted an offset-range test in the maze benchmark to verify the robustness of DAAC in faraway states. In particular, we use a DAAC policy trained in the setting of multi-map navigation without obstacles and with coordinates provided. When deploying the policy, we generate

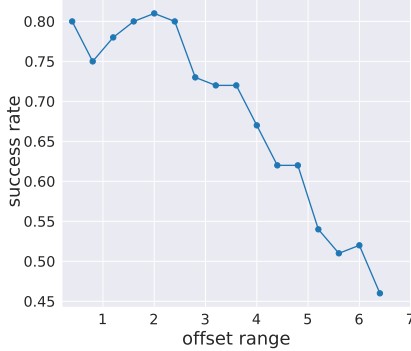

Figure 11: Illustration of DAAC in offset-range test. The X-axis is the offset range on the initial states, while the Y-axis shows the corresponding success rate of DAAC under this offset.

100 unseen maps and add positional offsets sampled from a uniform distribution to the initial states. We average the success rate under different ranges of offsets in Fig. 11.

In the maze environment, it's noteworthy that an offset range of initial states larger than could potentially make the agent be separated from the expert trajectory by a wall, which violates the property of 1-demo imitator availability in Prop. 2.2. Correspondingly, the results show that the policy sustains a respectable success rate within an offset of 2.4. After that, expanding the range

leads to a nearly linear decrease in success rate. The experiment demonstrated the agent's success in devising a well-performed policy for the scope of tasks with 1-demo imitator availability. Similar results can also be found in other experiments in which there is no exact match between the current and target state in these scenarios.

- In maze settings with obstacles (Fig. 25, 26, 29, and 30), we have observed the agent's remarkable ability to adaptively adjust its behavior when encountering obstacles.
- In robot manipulation tasks (Fig. 31-36), we present a showcase of the robotic arm's proficiency in following a trajectory while optimizing its operational efficiency. Moreover, in the corresponding video, which records rollouts generated by the DAAC policy, we can observe the simultaneous activation of multiple expert states through attention mechanisms when an exact match between the current state and the target state is lacking. The video can be found in the supplementary material.

In conclusion, within our solution scope, i.e., the problem with 1-demo imitator availability defined in Prop. 2.2, the imitator policy works well, even if the current states do not perfectly match the expert states. Besides, the policy still works to some degree in faraway states. However, we argue that without further information, assumptions, or prior knowledge, it is impossible to find a perfect imitator policy when the agent is inevitable to reach some faraway states. We leave the imitator learning problem in this setting as future work.

## G  MORE ABLATION STUDY RESULTS

- We have conducted an ablation study considering reward design in maze and complex robot manipulation tasks. In particular, for clipping term $\eta$ in $R_{\text{Itor}}$, we set $\eta$ as infinite value (DAAC-w/o clip), the results can be found in Fig. 12 and Fig. 13. The results show that directly removing the clipping function of $\eta$, which enhances the probabilities of ill-posedness led by the L2 distance, e.g., a wall obstructing the path between two states will have a small L2 distance, reduces the sample efficiency of DAAC, but the asymptotic performance is still similar, which demonstrates the robustness of DAAC to the ill-posedness of the L2 distance.

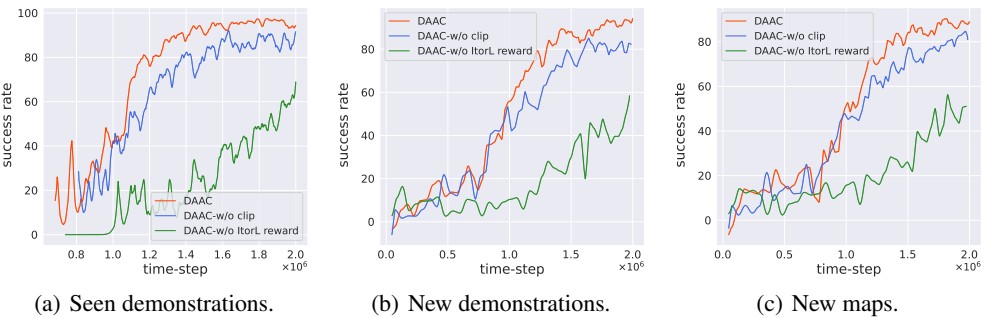

(a) Seen demonstrations.          (b) New demonstrations.          (c) New maps.

Figure 12: Learning curves of agents with varying reward settings in the demo-navigation benchmark. The task is the Multi-map imitation without obstacles and with coordinates.

## H  LEARNING CURVES

We list the learning curves in this section. Fig. 14, Fig. 15, and Fig. 16 show the learning curves in eight different navigation tasks respectively. Fig. 17 shows the BC loss of TRANS-BC, which is the mean squared error (MSE) loss between expert actions and agent actions. To ensure conciseness in our description, we employ the following abbreviations: "SM" for Single-Map, "MM" for Multi-Map, "Ob" for scenes with obstacles, "Non-Ob" for scenes without obstacles, "Co" for scenes with coordinates and "Non-Co" for scenes without coordinates.

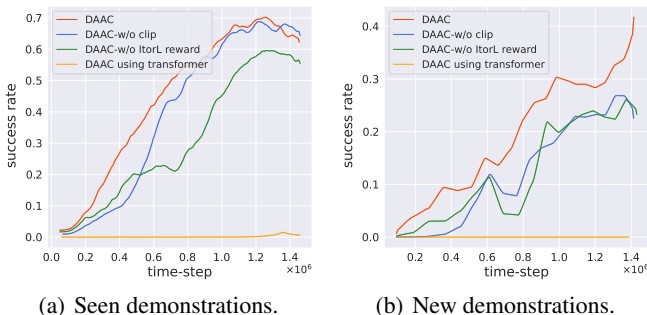

(a) Seen demonstrations.  (b) New demonstrations.

Figure 13: Learning curves of agents with varying training settings in the demo-manipulation environment. Note that the task is the Multi-task imitation which learns Grasping, Stacking, and Collecting simultaneously.

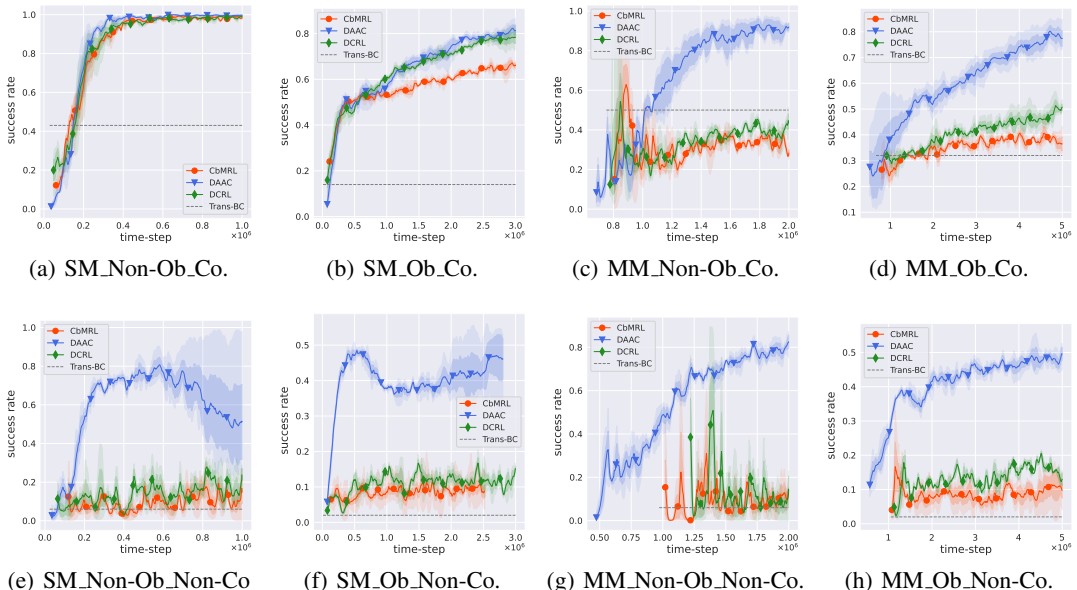

(a) SM_Non-Ob_Co.  (b) SM_Ob_Co.  (c) MM_Non-Ob_Co.  (d) MM_Ob_Co.

(e) SM_Non-Ob_Non-Co  (f) SM_Ob_Non-Co.  (g) MM_Non-Ob_Non-Co.  (h) MM_Ob_Non-Co.

Figure 14: Learning curves on demonstrations seen during the training.

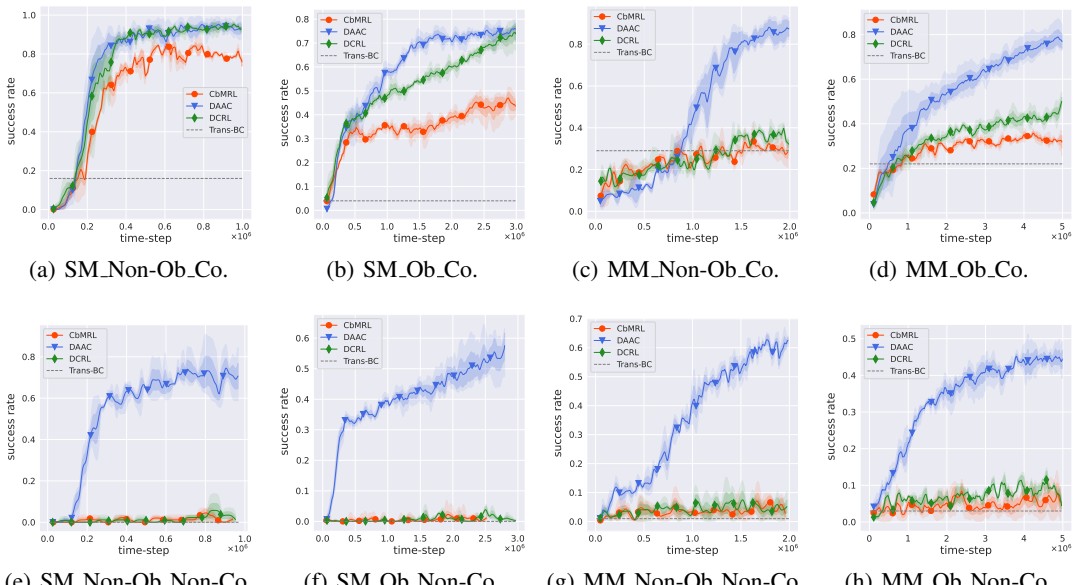

Figure 15: Learning curves on new demonstrations from seen maps.

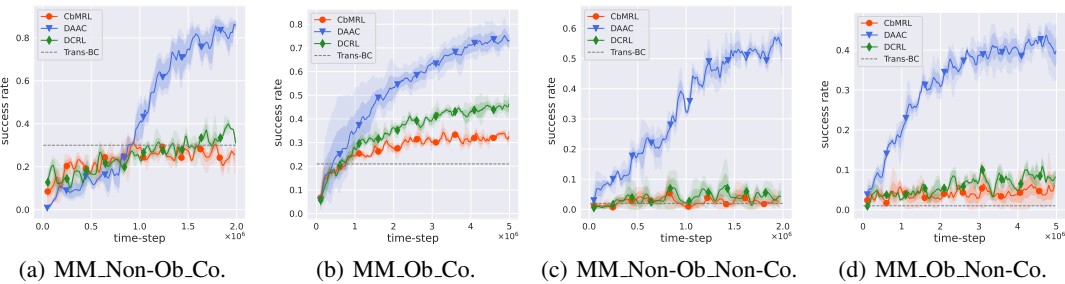

Figure 16: Learning curves on demonstrations collected from new maps.

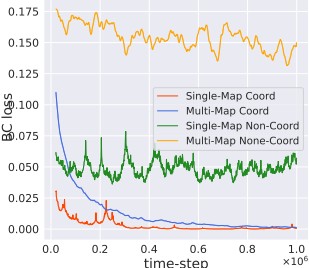

Figure 17: BC Loss of TRANS-BC. Note that both scenes with obstacles and tasks without obstacles use the same set of offline demonstrations, thus there are a total of four curves representing eight tasks.

# I  VISUALIZATION

In Fig. 18, we give more visualizations. In these tasks, the agent knows its coordinate and is required to imitate demonstrations collected on new maps without obstacles.

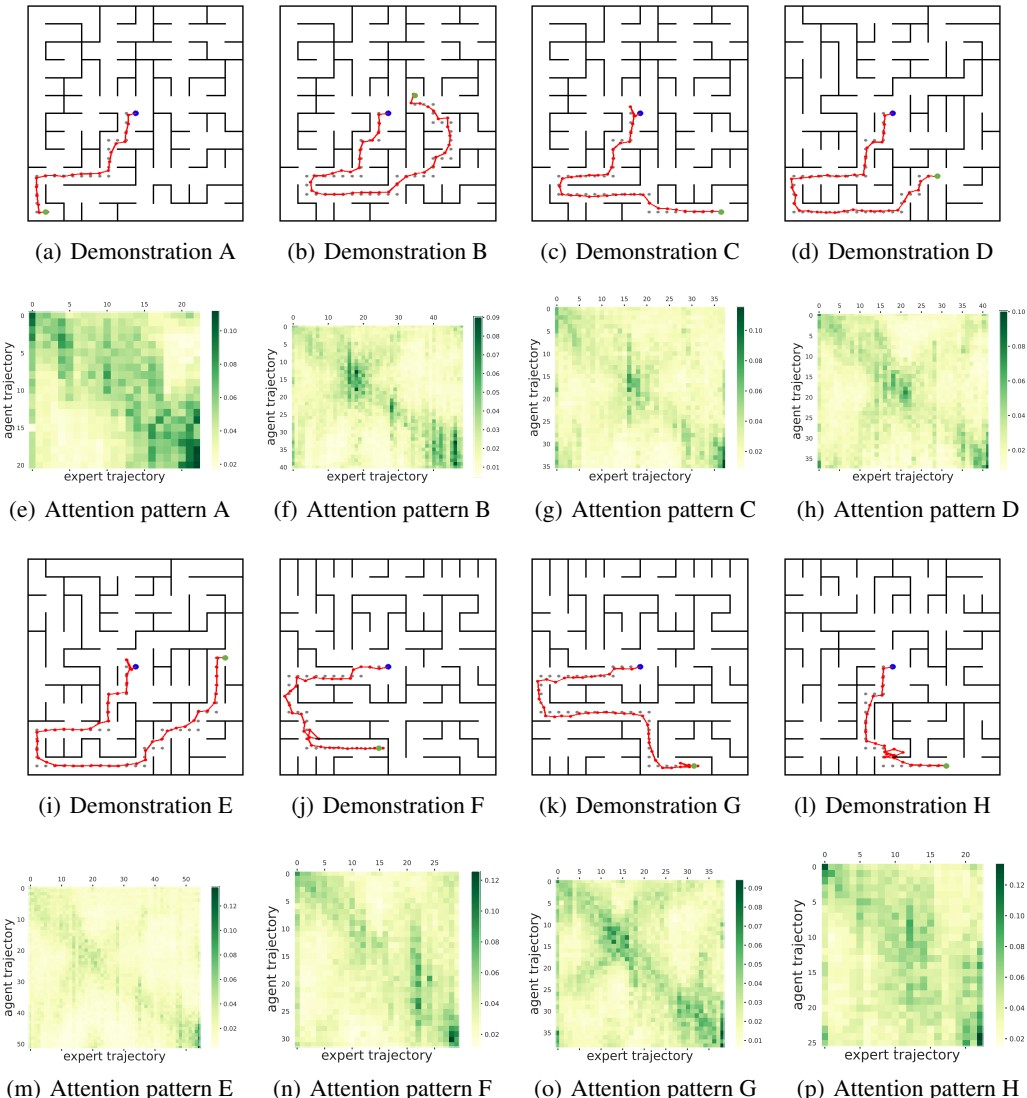

Figure 18: Illustrations of attention patterns in DAAC. In Fig.(e)-(h) and Fig.(m)-(p), the vertical axis of the attention score map corresponds to the trajectory of the agent, while the horizontal axis represents the trajectory of the expert. The intensity of color within a row indicates the level of attention allocated by the agent to the corresponding expert state. The imitator agent actively aligns with expert states by leveraging the matched state for decision-making, *with higher attention values predominantly concentrated along the diagonal*.

## J COMPARISONS OF TRAJECTORIES OF DAAC TRAINED BY DIFFERENT REWARDS

In Fig. 19, we provide visualized comparisons of trajectories of DAAC trained with our ItorL reward and without. In the tasks, the agent knows its coordinate and is required to imitate demonstrations collected on new maps without obstacles. Through random sampling of multiple tasks, we have observed that in specific scenarios, intelligent agents without ItorL reward tend to encounter wall collisions or deviate from the correct path, resulting in being lost. This behavior can potentially arise from their inclination to take shortcuts as a means to expedite reaching the goal.

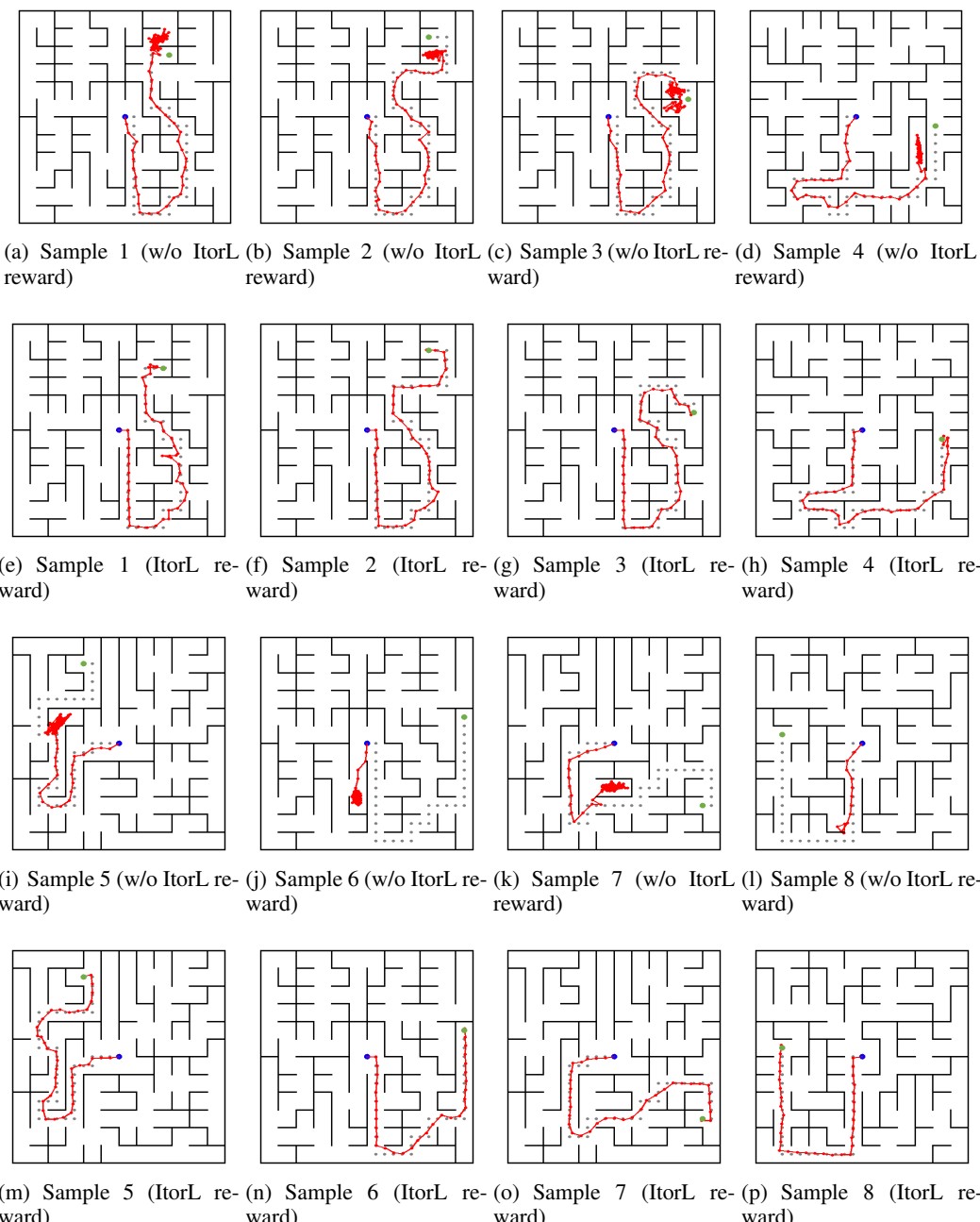

(a) Sample 1 (w/o ItorL reward) (b) Sample 2 (w/o ItorL reward) (c) Sample 3 (w/o ItorL reward) (d) Sample 4 (w/o ItorL reward)

(e) Sample 1 (ItorL reward) (f) Sample 2 (ItorL reward) (g) Sample 3 (ItorL reward) (h) Sample 4 (ItorL reward)

(i) Sample 5 (w/o ItorL reward) (j) Sample 6 (w/o ItorL reward) (k) Sample 7 (w/o ItorL reward) (l) Sample 8 (w/o ItorL reward)

(m) Sample 5 (ItorL reward) (n) Sample 6 (ItorL reward) (o) Sample 7 (ItorL reward) (p) Sample 8 (ItorL reward)

Figure 19: comparisons of trajectories of DAAC trained with or without the ItorL reward.

## K    THE DETAILS OF THE SCALING UP EXPERIMENTS

**Experiments on Different Demonstration Quantity**    To investigate the influence of demonstration quantity on model performance, we conducted experiments with four varying quantity settings: 60, 240, 960, and 2160. The results are in Fig. 20. We observed that fewer data leads to quicker initial learning speed and rapid performance improvement. However, a bottleneck emerges when aiming for generalization performance. As demonstration quantity increases, the learning task becomes more difficult, resulting in slower initial learning. Nonetheless, the final model exhibits notably superior performance on new demos and new maps compared to experiments conducted with fewer data.

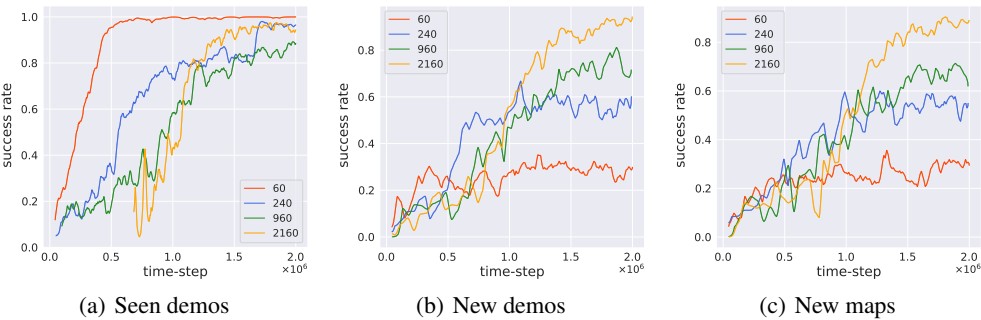

(a) Seen demos            (b) New demos            (c) New maps

Figure 20: Learning curves of agents with varying demonstration quantity. Note that the task is the Multi-map imitation without obstacles and with coordinates.

**Experiments on Different Model Parameters**    In this experiment, We focused on tuning $d_{model}$, nhead, $L_{encoder}$ and $L_{decoder}$ for both actor and critic to construct DAAC variants with different model parameters, where $d_{model}$ represents the desired number of features in the encoder/decoder inputs, nhead denotes the number of heads in the multi-head attention mechanism, $L_{encoder}$ represents the number of sub-encoder layers within the encoder, and $L_{decoder}$ refers to the number of sub-decoder layers within the decoder. The details of parameters selection are in Tab. 6, and the learning curves are in Fig. 21.

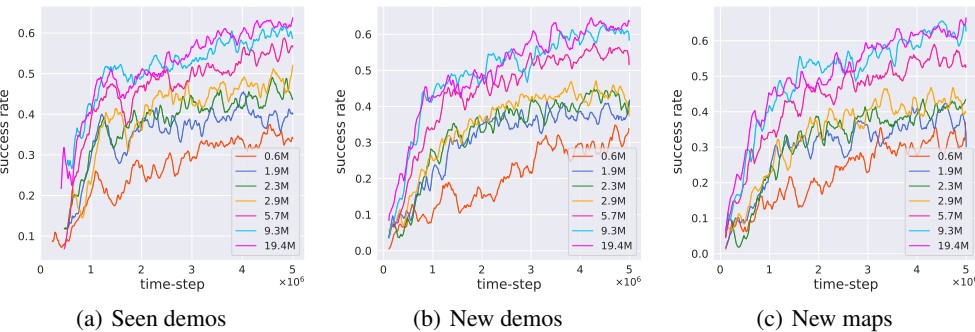

(a) Seen demos            (b) New demos            (c) New maps

Figure 21: Learning curves of agents with varying model sizes. Note that the task is the Multi-map imitation with obstacles and without coordinates.

Table 6: Architecture hyperparameters for DAAC variants with different model parameters.

| Total Parameters | Actor | | | | | Critic | | | | |
|---|---|---|---|---|---|---|---|---|---|---|
| | $d_{model}$ | nheads | $L_{encoder}$ | $L_{decoder}$ | parameters | $d_{model}$ | nheads | $L_{encoder}$ | $L_{decoder}$ | parameters |
| 0.6M | 64 | 16 | 3 | 3 | 0.4M | 64 | 16 | 2 | 2 | 0.2M |
| 1.9M | 64 | 32 | 3 | 6 | 0.6M | 128 | 16 | 4 | 4 | 1.3M |
| 2.3M | 96 | 32 | 3 | 6 | 1.0M | 128 | 16 | 4 | 4 | 1.3M |
| 2.9M | 128 | 64 | 3 | 6 | 1.6M | 128 | 16 | 4 | 4 | 1.3M |
| 5.7M | 192 | 16 | 3 | 6 | 3.1M | 192 | 16 | 4 | 4 | 2.6M |
| 9.3M | 256 | 16 | 3 | 6 | 5.1M | 256 | 16 | 4 | 4 | 4.2M |
| 19.4M | 384 | 16 | 3 | 6 | 10.7M | 384 | 16 | 4 | 4 | 8.7M |

## L    COMPARISONS OF TRAJECTORIES OF DIFFERENT LEARNING PARADIGMS

The demonstration input not only tells the agent which task to accomplish but the way to accomplish the task. To better illustrate that our DAAC method leverages information from demonstrations to understand the task and enables more efficient and better generalization ability learning by imitating the demonstrations, we provide visualized agent behaviors of different learning paradigms in Fig. 22. (a) TRANS-BC attempts to mimic demonstrations but lacks the ability to generalize to states not seen in the demonstrations, resulting in failure. (b) DCRL successfully completes the task but does not explicitly imitate the demonstrations. (c) In contrast, our DAAC method efficiently learns the task while imitating demonstrations. The results demonstrate the generalization ability learned by the imitator learning than other paradigms.

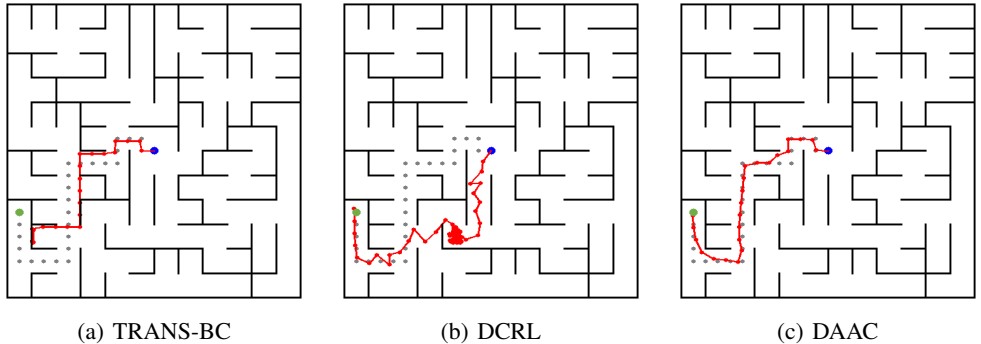

(a) TRANS-BC             (b) DCRL             (c) DAAC

Figure 22: Visualized trajectories of different learning paradigms.

# M MORE EXAMPLES OF DAAC TRAJECTORIES

## M.1 DAAC TRAJECTORIES IN MAZE ENVIRONMENTS

In Fig. 23-30, we show the trajectories generated by DAAC agents in all of the tasks.

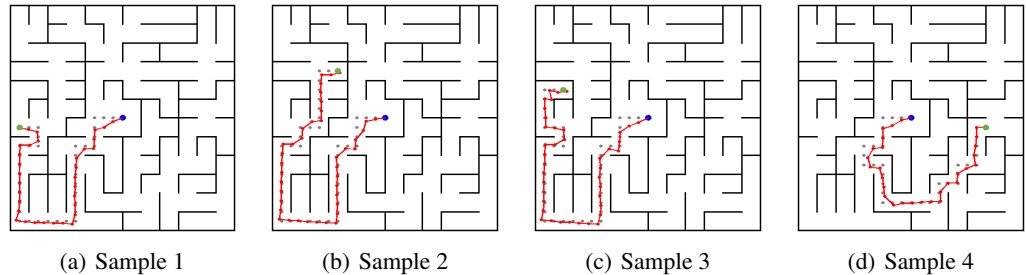

(a) Sample 1     (b) Sample 2     (c) Sample 3     (d) Sample 4

Figure 23: Illustration of trajectories generated by DAAC agents where the agents are trained on the single-map setting without obstacles and with coordinates provided.

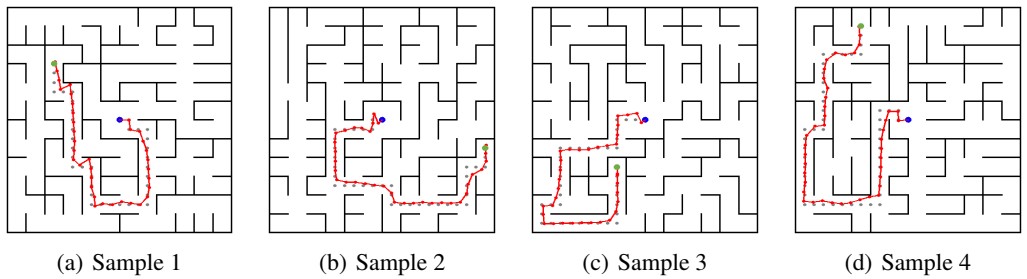

(a) Sample 1     (b) Sample 2     (c) Sample 3     (d) Sample 4

Figure 24: Illustration of trajectories generated by DAAC agents where the agents are trained on the multi-map setting without obstacles and with coordinates provided.

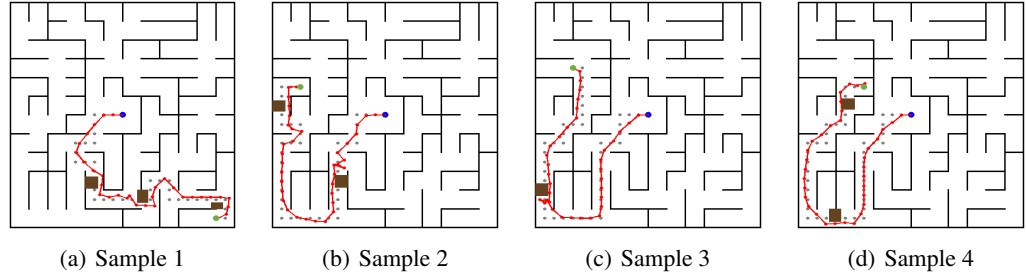

(a) Sample 1     (b) Sample 2     (c) Sample 3     (d) Sample 4

Figure 25: Illustration of trajectories generated by DAAC agents where the agents are trained on the single-map setting with obstacles and with coordinates provided.

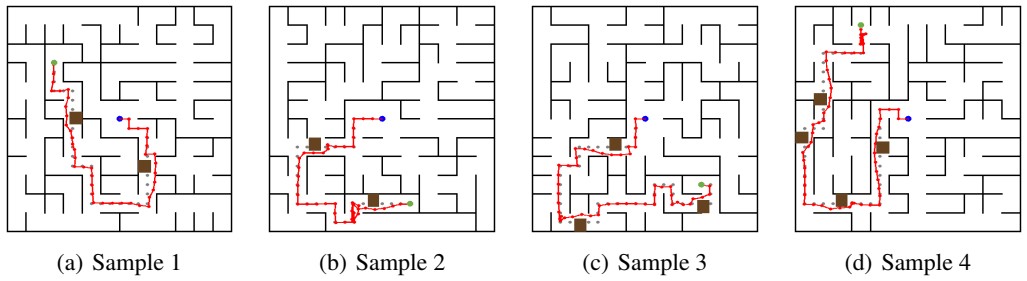

(a) Sample 1     (b) Sample 2     (c) Sample 3     (d) Sample 4

Figure 26: Illustration of trajectories generated by DAAC agents where the agents are trained on the multi-map setting with obstacles and with coordinates provided.

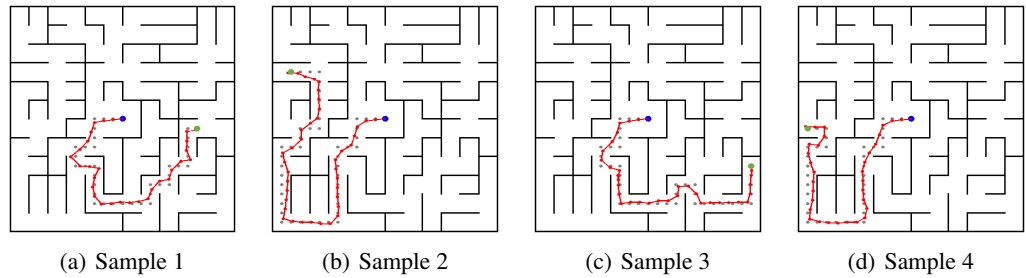

(a) Sample 1          (b) Sample 2          (c) Sample 3          (d) Sample 4

Figure 27: Illustration of trajectories generated by DAAC agents where the agents are trained on the single-map setting without obstacles and without coordinates provided.

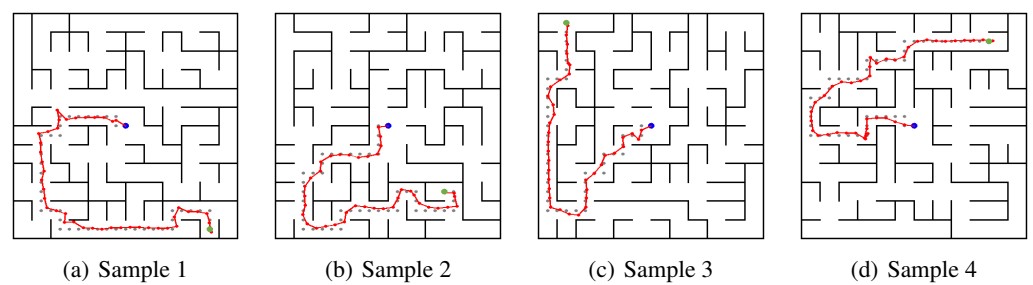

(a) Sample 1          (b) Sample 2          (c) Sample 3          (d) Sample 4

Figure 28: Illustration of trajectories generated by DAAC agents where the agents are trained on the multi-map setting without obstacles and without coordinates provided.

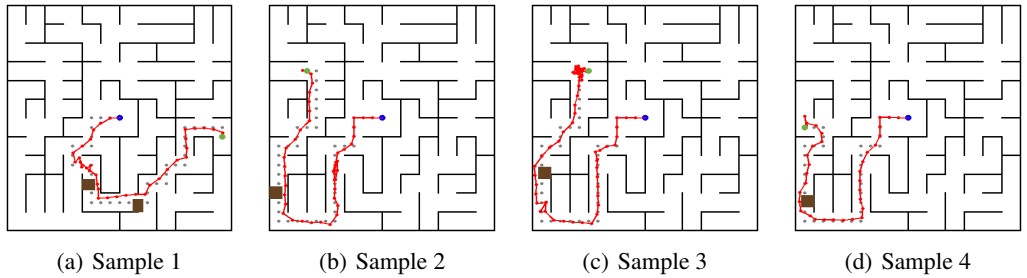

(a) Sample 1          (b) Sample 2          (c) Sample 3          (d) Sample 4

Figure 29: Illustration of trajectories generated by DAAC agents where the agents are trained on the single-map setting with obstacles and without coordinates provided.

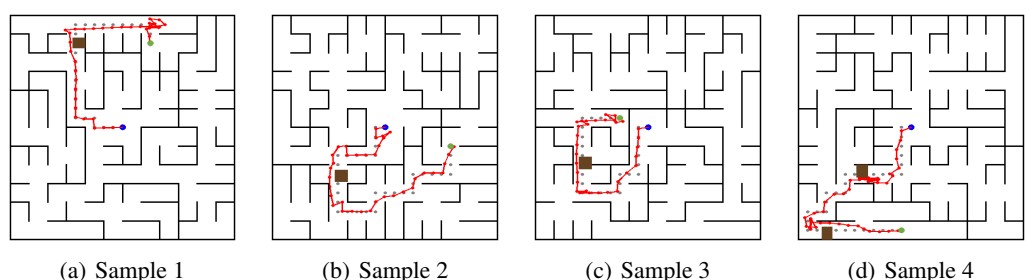

(a) Sample 1          (b) Sample 2          (c) Sample 3          (d) Sample 4

Figure 30: Illustration of trajectories generated by DAAC agents where the agents are trained on the multi-map setting with obstacles and without coordinates provided.

## M.2 DAAC TRAJECTORIES IN ROBOT MANIPULATION ENVIRONMENTS

In Fig. 31-36, we show the trajectories generated by DAAC agents in all of the tasks. In all of the figures, the red line represents the expert demonstrations collected by sequentially executing predefined heuristic primitives, *the lighter the latter*. The blue dots are event points denoting the agent trajectory, *the lighter the latter*. The event points generated by our method always distribute around the demonstration trajectories, which demonstrates that the agent actively matches expert states for decision making. **Besides, we recorded the corresponding videos in the supplementary material.**

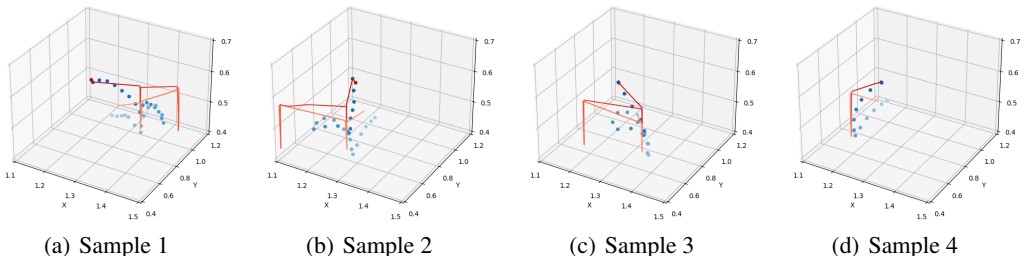

| (a) Sample 1 | (b) Sample 2 | (c) Sample 3 | (d) Sample 4 |

Figure 31: Illustration of trajectories generated by DAAC agents and corresponding demonstrations in object-grasping tasks. The agents are trained on object-grasping tasks and tested with unseen object-grasping demonstrations. The robot needs to grasp the target object without colliding with other objects.

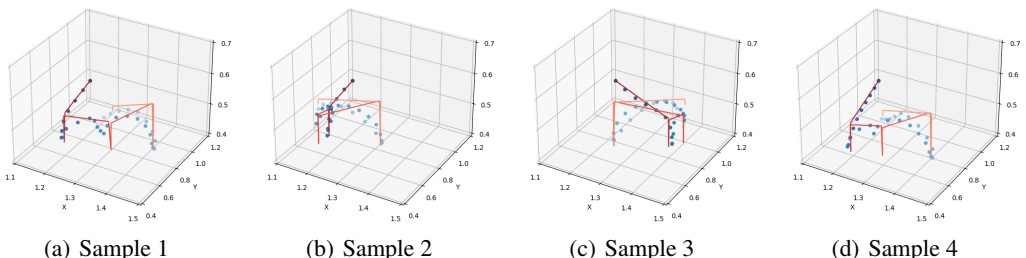

| (a) Sample 1 | (b) Sample 2 | (c) Sample 3 | (d) Sample 4 |

Figure 32: (a)-(d) Robot manipulation trajectories generated by DAAC agents. The agents are trained on object-stacking tasks and tested with unseen object-stacking demonstrations. The robot needs to stack three initially placed objects together.

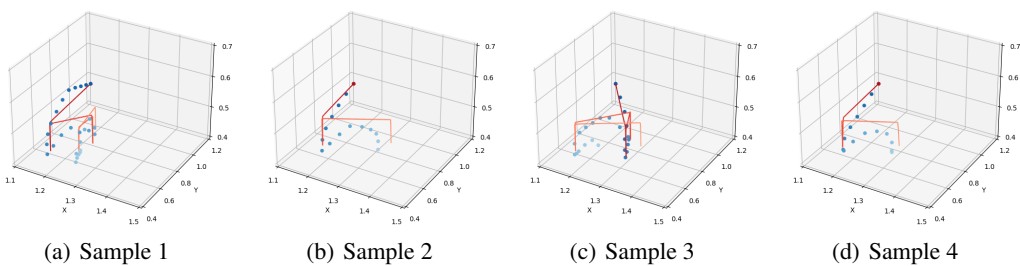

| (a) Sample 1 | (b) Sample 2 | (c) Sample 3 | (d) Sample 4 |

Figure 33: Illustration of trajectories generated by DAAC agents and corresponding demonstrations in object-collecting tasks. The agents are trained on object-collecting tasks and tested with unseen object-collecting demonstrations. The robot needs to collect all objects scattered over the desk and place them in the specified area (yellow).

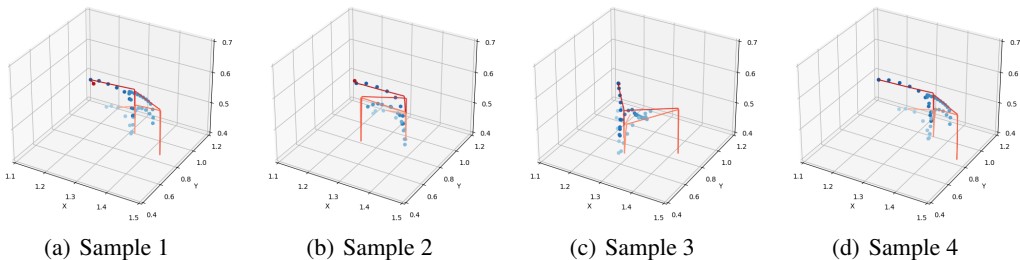

(a) Sample 1      (b) Sample 2      (c) Sample 3      (d) Sample 4

Figure 34: Illustration of trajectories generated by DAAC agents trained to imitate three types of manipulation tasks simultaneously. The agents are tested with unseen object-grasping demonstrations.

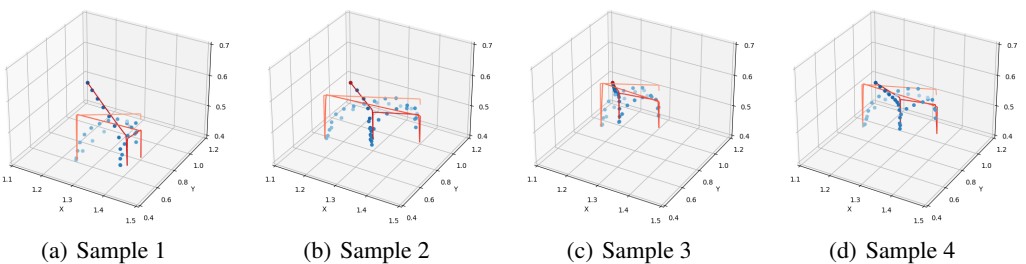

(a) Sample 1      (b) Sample 2      (c) Sample 3      (d) Sample 4

Figure 35: Illustration of trajectories generated by DAAC agents trained to imitate three types of manipulation tasks simultaneously. The agents are tested with unseen object-stacking demonstrations.

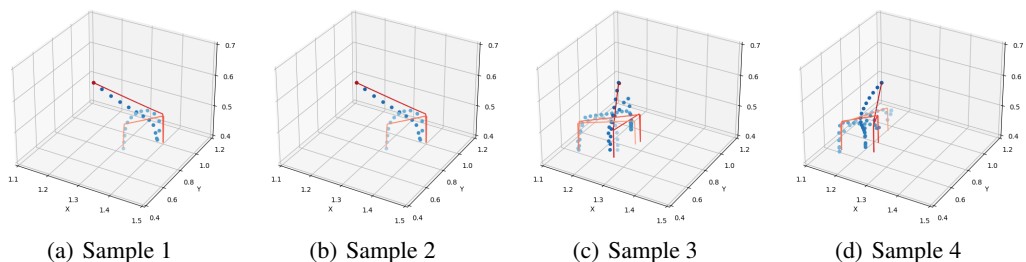

(a) Sample 1      (b) Sample 2      (c) Sample 3      (d) Sample 4

Figure 36: Illustration of trajectories generated by DAAC agents trained to imitate three types of manipulation tasks simultaneously. The agents are tested with unseen object-collecting demonstrations.

## N  SOCIETAL IMPACT

This work studies a new topic called imitator learning (ItorL), which aims to derive an imitator module that can *on-the-fly* reconstruct the imitation policies based on very *limited* expert demonstrations for different unseen tasks, without any extra adjustment. ItorL enables imitation in many real-world application applications where humans require performing various tasks out of the box, through very limited demonstrations of corresponding tasks. For example, for autonomous vehicles, we would like the vehicle to park in different parking lots directly (Ahn et al., 2022; Kümmerle et al., 2009) by presenting a human navigation trajectory; for robot manipulation, we aim for a robot arm to perform a variety of tasks directly (Dance et al., 2021; Yu et al., 2019) by just giving the corresponding correct operation demonstrations. Nonetheless, it is important to consider the ethical implications of

deploying these RL agents in real-world settings. Ensuring that these systems maintain transparency and accountability is of paramount importance.

APPENDIX REFERENCES

Jianyu Chen, Bodi Yuan, and Masayoshi Tomizuka. Deep imitation learning for autonomous driving in generic urban scenarios with enhanced safety. In *International Conference on Intelligent Robots and Systems*, pp. 2884–2890, 2019.

Xiong-Hui Chen, Yang Yu, Qingyang Li, Fan-Ming Luo, Zhiwei (Tony) Qin, Wenjie Shang, and Jieping Ye. Offline model-based adaptable policy learning. In *Advances in Neural Information Processing Systems*, pp. 8432–8443, 2021.

Christopher R. Dance, Julien Perez, and Théo Cachet. Demonstration-conditioned reinforcement learning for few-shot imitation. In Marina Meila and Tong Zhang (eds.), *International Conference on Machine Learning*, pp. 2376–2387, 2021.

Sudeep Dasari and Abhinav Gupta. Transformers for one-shot visual imitation. In *Conference on Robot Learning*, pp. 2071–2084, 2021.

Yan Duan, Marcin Andrychowicz, Bradly Stadie, OpenAI Jonathan Ho, Jonas Schneider, Ilya Sutskever, Pieter Abbeel, and Wojciech Zaremba. One-shot imitation learning. *Advances in Neural Information Processing Systems*, pp. 1087–1098, 2017.

Chelsea Finn, Pieter Abbeel, and Sergey Levine. Model-agnostic meta-learning for fast adaptation of deep networks. In *International Conference on Machine Learning*, pp. 1126–1135, 2017a.

Chelsea Finn, Tianhe Yu, Tianhao Zhang, Pieter Abbeel, and Sergey Levine. One-shot visual imitation learning via meta-learning. *Conference on Robot Learning*, pp. 357–368, 2017b.

Carlos Florensa, David Held, Xinyang Geng, and Pieter Abbeel. Automatic goal generation for reinforcement learning agents. In *International Conference on Machine Learning*, pp. 1514–1523, 2018.

Scott Fujimoto and Shixiang Shane Gu. A minimalist approach to offline reinforcement learning. In *Advances in Neural Information Processing Systems*, pp. 20132–20145, 2021.

Scott Fujimoto, Herke Hoof, and David Meger. Addressing function approximation error in actor-critic methods. In *International Conference on Machine Learning*, pp. 1587–1596, 2018.

Todd Hester, Matej Vecerík, Olivier Pietquin, Marc Lanctot, Tom Schaul, Bilal Piot, Dan Horgan, John Quan, Andrew Sendonaris, Ian Osband, Gabriel Dulac-Arnold, John P. Agapiou, Joel Z. Leibo, and Audrunas Gruslys. Deep q-learning from demonstrations. In *AAAI Conference on Artificial Intelligence*, pp. 3223–3230, 2018.

Jiayi Li, Tao Lu, Xiaoge Cao, Yinghao Cai, and Shuo Wang. Meta-imitation learning by watching video demonstrations. In *International Conference on Learning Representations*, 2021.

Fan-Ming Luo, Shengyi Jiang, Yang Yu, Zongzhang Zhang, and Yi-Feng Zhang. Adapt to environment sudden changes by learning a context sensitive policy. In *AAAI Conference on Artificial Intelligence*, pp. 7637–7646, 2022.

Zhao Mandi, Fangchen Liu, Kimin Lee, and Pieter Abbeel. Towards more generalizable one-shot visual imitation learning. In *International Conference on Robotics and Automation*, pp. 2434–2444, 2022.

Anusha Nagabandi, Ignasi Clavera, Simin Liu, Ronald S. Fearing, Pieter Abbeel, Sergey Levine, and Chelsea Finn. Learning to adapt in dynamic, real-world environments through meta-reinforcement learning. In *International Conference on Learning Representations*, 2019.

Suraj Nair, Silvio Savarese, and Chelsea Finn. Goal-aware prediction: Learning to model what matters. In *International Conference on Machine Learning*, pp. 7207–7219, 2020.

Andrew Y. Ng and Stuart Russell. Algorithms for inverse reinforcement learning. *International Conference on Machine Learning*, pp. 663–670, 2000.

OpenAI, Ilge Akkaya, Marcin Andrychowicz, Maciek Chociej, Mateusz Litwin, Bob McGrew, Arthur Petron, Alex Paino, Matthias Plappert, Glenn Powell, Raphael Ribas, Jonas Schneider, Nikolas Tezak, Jerry Tworek, Peter Welinder, Lilian Weng, Qiming Yuan, Wojciech Zaremba, and Lei Zhang. Solving Rubik's cube with a robot hand. *arXiv preprint arXiv:1910.07113*, 2019.

Yunpeng Pan, Ching-An Cheng, Kamil Saigol, Keuntaek Lee, Xinyan Yan, Evangelos A. Theodorou, and Byron Boots. Agile autonomous driving using end-to-end deep imitation learning. In *Robotics: Science and Systems*, 2018.

Xue Bin Peng, Marcin Andrychowicz, Wojciech Zaremba, and Pieter Abbeel. Sim-to-Real transfer of robotic control with dynamics randomization. In *International Conference on Robotics and Automation*, pp. 1–8, 2018.

Dean Pomerleau. Efficient training of artificial neural networks for autonomous navigation. *Neural Computation*, pp. 88–97, 1991.

Aravind Rajeswaran, Vikash Kumar, Abhishek Gupta, Giulia Vezzani, John Schulman, Emanuel Todorov, and Sergey Levine. Learning complex dexterous manipulation with deep reinforcement learning and demonstrations. In *Robotics: Science and Systems*, 2018.

Kate Rakelly, Aurick Zhou, Chelsea Finn, Sergey Levine, and Deirdre Quillen. Efficient off-policy meta-reinforcement learning via probabilistic context variables. In *International Conference on Machine Learning*, pp. 5331–5340, 2019.

Stéphane Ross and Drew Bagnell. Efficient reductions for imitation learning. In *International Conference on Artificial Intelligence and Statistics*, pp. 661–668, 2010.

Stéphane Ross, Geoffrey J. Gordon, and Drew Bagnell. A reduction of imitation learning and structured prediction to no-regret online learning. In *International Conference on Artificial Intelligence and Statistics*, pp. 627–635, 2011.

Ashish Vaswani, Noam Shazeer, Niki Parmar, Jakob Uszkoreit, Llion Jones, Aidan N. Gomez, Lukasz Kaiser, and Illia Polosukhin. Attention is all you need. In *Advances in Neural Information Processing Systems*, pp. 5998–6008, 2017.

Fan Xie, Alexander Chowdhury, M. Clara De Paolis Kaluza, Linfeng Zhao, Lawson L. S. Wong, and Rose Yu. Deep imitation learning for bimanual robotic manipulation. In *Advances in Neural Information Processing Systems*, pp. 2327–2337, 2020.

Jia-Fong Yeh, Chi-Ming Chung, Hung-Ting Su, Yi-Ting Chen, and Winston H. Hsu. Stage conscious attention network (SCAN): A demonstration-conditioned policy for few-shot imitation. In *AAAI Conference on Artificial Intelligence*, pp. 8866–8873, 2022.

Tianhe Yu, Chelsea Finn, Annie Xie, Sudeep Dasari, Tianhao Zhang, Pieter Abbeel, and Sergey Levine. One-shot imitation from observing humans via domain-adaptive meta-learning. *arXiv preprint arXiv:1802.01557*, 2018.

