# OpenReview forum: "Learn to Achieve Out-of-the-Box Imitation Ability from Only One Demonstration"
_ICLR.cc/2024/Conference — Submitted to ICLR 2024_

### Official Review · Reviewer_7pta · 2023-10-30

**Soundness:** 3 good
**Presentation:** 2 fair
**Contribution:** 2 fair
**Rating:** 6
**Confidence:** 3

**Summary:**

Imitation Learning (IL) enables agents to mimic expert behaviors. Most previous IL techniques focus on precisely imitatiing one policy through mass demonstrations. However, in many applications, what humans require is the ability to perform various tasks directly through a few demonstrations of corresponding tasks, where *the agent would meet many unexpected changes when deployed*. In this scenario, the agent is expected to not only imitate the demonstration but also adapt to unforeseen environmental changes. This motivates the authors to propose a new topic called imitator learning (ItorL), which aims to derive an imitator module that can *on-the-fly* reconstruct the imitation policies based on very limited expert demonstrations for different unseen tasks, without extra adjustment. In this work, the authors focus on imitator learning based on only one expert demonstration. To solve ItorL, the authors propose Demo-Attention Actor-Critic (DAAC), which integrates IL into a reinforcement-learning paradigm that can regularize policies' behaviors in unexpected situations. Besides, for autonomous imitation policy building, they design a demonstration-based attention architecture for imitator policy that can effectively output imitated actions by adaptively tracing the suitable states in demonstrations. They develop a new navigation benchmark and a robot environment for ItorL and show that DAAC outperforms previous imitation methods *with large margins* both on seen and unseen tasks.

**Strengths:**

1. The paper is generally well written and easy to follow.
2. The experiments in this paper are sufficient and convincing to reflect the advantages of the proposed method.
3. The imitator learning proposed in this paper is interesting and meets the requirements of practical applications better compared with conventional imitation learning. Especially, I believe the *out-of-box* adapting ability which no more needs further fine-tuning is exactly what we need to utilizing reinforcement learning in practice. Therefore, I think this paper can encourage more exploration works along this meaningful direction.

**Weaknesses:**

1. Some figures need to be improved, like Figure 4. Some texts in this figure exceed the maze boundaries and some texts overlap with the walls in the maze. I suggest using the figure legend to replace the texts in the figure.
2. It will be better if some theoretical guarantees for the imtation ability of the propose method can be provided.
3. The imitator reward design in Equ. 1 is heuristic and may lack the enough abillity to generalize to more complex or high-dimensional environments.
4. The authors are expected to elaborate more on the difference between the imitator learning and the meta reinforcement learning.

**Questions:**

1. The authors are encouraged to discuss more on if the handcrafted imitator reward design will limit the ability of the imitator module, though this module design itself is very creative.
2. The authores are expected to elaborate more on the ability of the proposed method on high-dimensional input environments, like Atari Games.
3. It will be much better if the authors can bring more thoughts/ideas on how to design an unified imitator reward or an automatic imitator reward producer according to specific environments.

---

> ### Author Response · Authors · 2023-11-20
> **Official resposne (part 1)**
>
> Thanks for the insightful suggestions on our paper.   We response the questions and concerns as follows:
>
> ### **Q1:  It will be better if some theoretical guarantees for the imitation ability of the propose method can be provided.**
>
> We concur with the importance of a solid theoretical foundation for advancing our method. In Appendix. B, we have conducted a detailed analysis of each component of our imitator reward. The theoretically validated imitation reward [1] $R_{IL}$  achieves imitation learning through reinforcement learning. A clipping term η to make the imitation rewards based on the too-far state pairs invalidated to avoid the potential misleading. A reweighting item $1/ \exp(d( ̄s, s)^2)$ is to avoid the agent overly penalizing for not strictly following the expert action when its current state is far from the demonstration state rewards on action matching. We add the ending reward to make the agent focus more on task completion rather than repeatedly collecting $R_{IL}$ rewards, and α rescales the ending rewards to make sure the ending reward is essential. R_Itor serves as a crucial signal by providing a dense reward, enabling the agent to closely follow the demonstrations and accomplish tasks effectively during the early stages. We will extend the intriguing theoretical analysis of $R_{Itor}$ in the future.
>
> [1] Imitation Learning by Reinforcement Learning. ICLR 2022.
>
> ### **Q2: The imitator reward design in Equ. 1 is heuristic and may lack the enough ability to generalize to more complex or high-dimensional environments.**
>
> We tested our method on the MetaWorld environment, where the robot's state comprises 39 dimensions, significantly higher than 10 dimensions in the maze environment. We compared our method, DAAC, with other baselines, and the final success rates for each method are as follows:
>
> | Time step | DAAC | DCRL | CbMRL | TRANS-BC |
> | --- | --- | --- | --- | --- |
> | 0.5M | 0.980 | 0.240 | 0.640 | 0.160 |
> | 1.5M |  1.00 | 0.925 | 1.00 | 0.084 |
>
> Our approach consistently outperforms all baseline methods and converges at 0.5 million timesteps. Although DCRL and CbMRL eventually accomplish the task in later stages, they require 1.5 million timesteps, resulting in three times the computational cost compared to our approach.
>
> This demonstrates the good generalization ability of our method.
>
> ### **Q3: The authors are expected to elaborate more on the difference between the imitator learning and the meta reinforcement learning.**
>
> Thanks for the suggestions.  In the new version, we revised the related work section by moving the discussion with meta-IL studies to the main body. Please refer to Section 3 in the new submission.
>
> ### **Q4: The authors are encouraged to discuss more on if the handcrafted imitator reward design will limit the ability of the imitator module, though this module design itself is very creative.**
>
> We understand the reviewer's concern about the heuristically designed imitator reward function. In the discussion within Appendix B, we identify three types of ill-posed scenarios: directly approaching the expert state, leading to entry into an unsafe state (State A); blindly mimicking expert actions when far from demonstrations (State B); and consistently remaining in the vicinity of expert states (State C). While we offer several techniques to ensure that $R_{\rm Itor}$ provides reasonable rewards within the state-action space, it is still possible to encounter ill-posed rewards in certain corner cases without additional task-specific information. Furthermore, from a theoretical perspective, we present evidence in Appendix B indicating that ill-posedness stemming from the L2 distance may slow down the convergence speed.
>
> Despite these unavoidable corner cases, we want to emphasize the robustness of our imitator reward function. As discussed in Appendix B, we have pointed out that the inclusion of ending rewards addresses certain ill-posed aspects of the reward function, affecting only the initial convergence speed and not the final performance of the policy. This is further supported by the ablation experiments in Appendix G.
>
> ### **Q5: The authores are expected to elaborate more on the ability of the proposed method on high-dimensional input environments**
>
> We concur that vision-based RL tasks, especially those like Atari Games, are significant. However, as outlined in Section 3, our algorithm is explicitly designed for MDP scenarios, not POMDPs, which many vision-based RL tasks manifest as. In POMDP settings, we cannot determine the optimal action based on single-step observation, which invalidates the attention mechanism in DA architecture. We have given a formal discussion about this limitation in Section 6 .

---

> ### Author Response · Authors · 2023-11-20
> **Official resposne (part 2)**
>
> ### **Q6: It will be much better if the authors can bring more thoughts/ideas on how to design an unified imitator reward or an automatic imitator reward producer according to specific environments.**
>
> Given the expert demonstrations, we can employ Inverse Reinforcement Learning (IRL) to reverse-engineer the reward function of the task automatically, facilitating imitation of the demonstrated behavior. We appreciate the reviewer's interest in this aspect and will consider their feedback in our future developments.
>
> ### **Q7: Some figures need to be improved, like Figure 4. Some texts in this figure exceed the maze boundaries and some texts overlap with the walls in the maze.**
>
> Thank you for your valuable feedback.  Following your suggestions, we have made necessary adjustments to enhance clarity and readability.

---

> > ### Comment · Reviewer_7pta · 2023-11-23
> >
> > Thanks for your detailed response. Most of my concerns have been alleviated. In Q2, when talking about the more complex or high-dimensional environments, I also encourage the authors to explore the effectiveness and generalization ability of the proposed method in vision-input environments, like Atari Games. Overall, I still appreciate the authors' effort on providing more empirical results. So I prefer to maintain my previous score.

---

### Official Review · Reviewer_nbDi · 2023-11-01

**Soundness:** 3 good
**Presentation:** 2 fair
**Contribution:** 3 good
**Rating:** 5
**Confidence:** 3

**Summary:**

This paper poses the "Imitator Learning" problem, wherein the goal is to learn a policy that can reconstruct expert policies given a small number of demonstrations on unseen tasks. The "Demo-Attention Actor-Critic" (DAAC) algorithm is proposed, which uses uses context-based meta-RL with an imitator reward. The imitator reward is based on a heuristic measure of similarity to expert state-action pairs. The proposed architecture uses an attention mechanism which computes attention scores between the current state and the states in the demonstrations, and it is argued that this provides implicit regularization of the policy behavior. Results are shown in maze environments where global map information is not known, as well as state-based robot manipulation tasks.

**Strengths:**

- This paper addresses an interesting problem coined "imitator learning" in which an agent is trained on an offline dataset of demonstrations for various tasks that have some shared structure. At test time, the agent receives a small number of demonstrations and the goal is for the agents to mimic the experts without fine-tuning. The problem/approach includes aspects similar to meta-RL, meta-IL, online imitation learning, and IRL but the precise problem statement seems to be original.
- The authors provide ablations of their proposed reward function and architecture and show that both components are useful for performance. The authors also include results on scaling trends.
- The paper has a thorough account of implementation and environment details in the appendix which is helpful for reproducibility.

**Weaknesses:**

- The assumption that tasks must be in a tracebackable MDP set seems to be a bit limiting. It would be useful to provide examples of what types of environments this assumption holds in and what the limitations are of this assumption. For example, it seems that this assumption may not hold in a variety of robotic manipulation tasks where some subtasks may be irreversible.

- The proposed imitator reward (a) relies on a distance function between states (which could be hard to scale meaningfully to high-dimensional observations) and (b) has several heuristic components and hyperparameters to tune.

- Related to the above point: One concern is that the method (including architecture and the reward mechanism) might not generalize well to other tasks, since there are multiple moving parts and hyperparameters to tune. Have the authors looked into testing on existing meta-RL benchmark suites (e.g. Meta-World)?

- The demonstration-based attention architecture relies on performing attention between the demonstration states and the current state. As the authors briefly mention, there are issues with this approach when the demonstration states are faraway from the current state. It does seems quite possible that faraway states may be encountered (e.g. due to compounding error, exploration, etc.)

- I think the paper could be organized a bit better overall, and a variety of important details are left to the Appendix. For example, it would be good to include more discussion of related work in the main paper, especially comparisons to online imitation, few-shot imitation, meta-RL, meta-imitation learning etc. when defining the imitator problem.

**Questions:**

I have included some questions above in the Weaknesses section and additional questions below.

- Could the authors please elaborate on the following claim: "we also observe that just regarding demonstrations as context vectors are inefficient in fully mining the knowledge implied in these data efficiently, e.g., the demonstration sequence not only tells the agent which task to accomplish but the way to accomplish the task" --> Could the authors elaborate on the evidence for this and what exactly the problem is? Further, could the authors provide evidence that using the attention network is an effective policy regularizer? It seems that this regularization term would also be pushing the behavior to the "way the task was accomplished" according to the expert.

- Each iteration of the algorithm requires running SAC to update the task-information extractor/context-based policy, which could be costly. Could the authors please provide details on how costly the algorithm is to run?

- As the authors mention, DAAC's performance is limited in scenarios where the ground truth state is not available. Could the authors describe where the algorithm fails in partially-observed scenarios?


Minor/Typos:
- p.3: "ItorLin"
- p. 34: "ItorLEnables"
- p. 34: "where humans require is performing"
- p. 6 "data-processing pipeline inner the policy network"
- R is defined twice
- Fig. (b) defines the Imitator learning as receiving a "few demos" in the target task, while the text states "we require the imitator policy to use only one demonstration"

---

> ### Author Response · Authors · 2023-11-20
> **Official response (part 1)**
>
> Thank you for your valuable feedback on our paper.  We response the questions and concerns as follows:
> ### **Q1-1: The assumption that tasks must be in a tracebackable MDP set seems to be a bit limiting. It would be useful to provide examples of what types of environments this assumption holds in and what the limitations are of this assumption.**
>
> Thank you for your insightful feedback regarding the mentioned assumption. We agree that our initial manuscript did not provide an intuitive explanation for this assumption.  To address this, we have revised Sections 2.2 and 4.2, as well as the Appendix, to offer a clearer understanding of this concept.
>
> The cornerstone of Proposition 2.2  is the unified goal-conditioned policy $\beta$, as defined in Definition 2.1. The fundamental idea behind $\beta$ is to enable the agent to revert to states in the demonstrations, irrespective of the unseen state that the task or any unforeseen changes in the environment might lead the agent to.
>
> This assumption is practical in various applications. For instance, in navigation tasks like parking, an agent might encounter unexpected obstacles or pedestrians not present in the demonstrations. The consistent behavior in such situations involves executing avoidance maneuvers until reaching a safe state, followed by tracing back to the following states. Generally speaking, if the policy $\beta$ exists, it allows the agent to follow the demonstrations and achieve the goal by continually tracing a reachable successor state $g \in \tau_{\omega}$, guided by $\beta$ until the goal state is reached, **which enables imitating without observing all expert actions**.  Similarly, for robot manipulation tasks, whatever disturbance a robot arm might encounter, if we always have $\beta$ to reach some of the successor states in the demonstrations, we can reach the goal by repeatedly calling $\beta$ with suitable goals. Briefly note that this consistent behavior is not required for all states. As delineated in Def. 2.1, focusing on states in $\tau_{\omega}$ and $R(d_0)$ is sufficient to establish the feasibility of a 1-demo imitator, as fully discussed in App.A.
>
> ###  **Q1-2: For example, it seems that this assumption may not hold in a variety of robotic manipulation tasks where some subtasks may be irreversible**
>
> If there are some subtasks may be irreversible for robotic manipulation, in the process of RL training on the task reward, the agent will be trained to complete these subtasks as precise as possible. This is also one of the feature in ItorL than IL. However, we also argue that if the agent is **inevitable to** reach some faraway states where the way back to the demonstrations is not existed or depends on some information hidden in the tasks we meet, without further information, assumptions, or prior knowledge, it is impossible for an algorithm to find a perfect imitator policy. We leave the imitator learning problem in this setting as future work. However, in practice, the problem can always be solved by adding enough information to the agent for decision-making.
>
> In conclusion, your feedback has prompted a thorough revision of our manuscript in Section 2.2 for better clarity on the concept of a tracebackable MDP set. We believe these changes will make our assumptions and their implications more understandable.
>
> ### **Q2-1: One concern is that the method (including architecture and the reward mechanism) might not generalize well to other tasks, since there are multiple moving parts and hyperparameters to tune.**
>
> Thank you for your feedback. While our method involves multiple moving parts and hyperparameters, we want to emphasize its inherent robustness of our method which primarily requires modifications to parameters in the reward function, namely η and α (Refer to Table 3 for details). The clipping term η to make the imitation rewards based on the too-far state pairs invalidated to avoid the potentially misleading and α rescales the ending rewards to make sure the ending reward is essential.
>
> ### **Q2-2: Have the authors looked into testing on existing meta-RL benchmark suites (e.g. Meta-World)**
>
> Following your suggestion, we deliberately refrained from adjusting any parameters and transferred our algorithm to the pick-and-place environment in Meta-World. The final success rates for each method at different timesteps are as follows:
>
> | Time step | DAAC | DCRL | CbMRL | TRANS-BC |
> | --- | --- | --- | --- | --- |
> | 0.5M | 0.980 | 0.240 | 0.640 | 0.160 |
> | 1.5M |  1.00 | 0.925 | 1.00 | 0.084 |
>
> The results demonstrate strong performance, surpassing all baseline algorithms. Note that our method converges at 0.4 million timesteps. While DCRL and CbMRL finally complete the task in later stages, it takes them 1.5 million timesteps, resulting in **three times the computational effort** compared to our approach. We believe this showcases the robustness and effectiveness of our approach without extensive parameter adjustments

---

> ### Author Response · Authors · 2023-11-20
> **Official response (part 2)**
>
> ### **Q3: As the authors briefly mention, there are issues with this approach when the demonstration states are faraway from the current state. It does seems quite possible that faraway states may be encountered (e.g. due to compounding error, exploration, etc.)**
>
> Thank you for highlighting the issue regarding the potential of encountering faraway states in our approach. We would like to clarify that **it is ok to encounter too faraway states as long as the optimal policy can avoid it**. We take the compounding error and exploration problem to demonstrate the point:
>
> 1. **Compounding error**: the compounding error issue will be crucial in standard offline IL settings, which has also been discussed in Section 3 (revised version). However, in DAAC, **we learn to imitate via reinforcement learning, which naturally solves the compounding error issue, as the imitator policy is asked to optimize task performance**: In the learning process, due to the stochastic exploration of the imitator policy, the agent will face numerous unseen states. To better achieve the goals, the policy gradient should guide the policy to adjust the agent's trajectory in these states, thereby correcting deviations. To illustrate this, we have added visualizations of trajectories learned through ItorL and IL in our revised manuscript, highlighting the differences in policy behavior under both paradigms. (refer to Appendix L).
> 2. **Exploration**: While exploration might initially lead the agent to faraway states, our imitator reward design (referenced in Eq. (1)) encourages the policy to roughly align with the demonstrations. Once the imitator policy becomes proficient at tracing the demonstrations, it naturally avoids actively reaching these faraway states.
>
> We suspect the confusion may have arisen from the imprecise statements in the last paragraph of Section 4.2 and Appendix F. In response, we have thoroughly revised these sections for clearer articulation. We hope that the modifications in our revised manuscript now effectively address your concerns regarding the handling of faraway states in our approach.
>
> ### **Q4: I think the paper could be organized a bit better overall, and a variety of important details are left to the Appendix. For example, it would be good to include more discussion of related work in the main paper, especially comparisons to online imitation, few-shot imitation, meta-RL, meta-imitation learning etc. when defining the imitator problem.**
>
> Thanks for the suggestions. We also realized that the readers’ major concerns on the related work are about the differences compared with the similar setting of IL instead of the related implementations. In the new version, we revised the related work section by moving the discussion with meta-IL studies to the main body. Please refer to Section 3 in the new submission.
>
> ### **Q5:Could the authors elaborate on the evidence for this and what exactly the problem is?**
>
> Thank you for the insightful question. Our DAAC method leverages information from demonstrations to understand the task, which also enables more efficient and better generalization ability learning by imitating the demonstrations.
>
> In the revised version, we provide visual results of agent behaviors in Figure 22 in Appendix L. Trans-BC attempts to mimic demonstrations but lacks the ability to generalize to states not seen in the demonstrations, resulting in failure. DCRL successfully completes the task but does not explicitly imitate the demonstrations. In contrast, our approach efficiently learns the task while imitating demonstrations. The results demonstrate the generalization ability learned by the imitator learning than other paradigms.
>
> The effectiveness of our method is further evident in the learning curves presented in Figures 14-16, demonstrating that our approach outperforms all baseline algorithms in terms of learning speed.

---

> ### Author Response · Authors · 2023-11-20
> **Official resposne (part 3)**
>
> ### **Q6:could the authors provide evidence that using the attention network is an effective policy regularizer?  It seems that this regularization term would also be pushing the behavior to the "way the task was accomplished" according to the expert.**
>
> Thank you for your insightful query regarding the efficacy of our proposed attention network as a policy regularizer in our research.
>
> To address your question, it is imperative to understand that the regularizer in our context is essentially *the inductive bias introduced by the neural network architecture*. Inductive bias, in machine learning, refers to the set of assumptions that a model naturally adapts to generalize from training data to unseen scenarios. For instance, in Convolutional Neural Networks (CNNs), the inductive bias is in the form of local connectivity and shared weights, which are effective in recognizing patterns in spatial data. Similarly, Recurrent Neural Networks (RNNs) exhibit a temporal inductive bias, making them suitable for sequential data processing due to their ability to maintain state over time.
>
> In the context of our work, we incorporate a 'demo-attention' architecture as an inductive bias. This design choice is grounded in our prior knowledge that the imitation process in learning tasks is typically comprised of two fundamental phases: (1) determining the state to follow, and (2) deciding the action to take. Our demo-attention architecture is explicitly designed to mimic this two-phase process in the policy network.  By embedding this structured approach directly into the architecture of the policy network, we ensure that the policies learned are not only tailored to the training tasks but also possess inherent flexibility. This flexibility allows the policies to better generalize to unseen tasks and demonstrations, which is a core objective in Imitator Learning. *The demo-attention mechanism, therefore, acts as a regularizer not by constraining the policy but by guiding it to learn in a manner that is consistent with how expert demonstrations are typically structured.*
>
> We believe that this architectural inductive bias is a significant step towards more generalizable and robust policy learning in imitation tasks. The effectiveness of this approach is further substantiated by our experimental results, which demonstrate enhanced performance and adaptability in varied imitation scenarios.
>
> In response, we add a sentence in the last paragraph of Section 4.2 to explicitly reveal the relation between the regularizer and the inductive bias. We hope this explanation clarifies the rationale behind our design choice and its effectiveness as a policy regularizer.
>
> ### **Q7: Could the authors please provide details on how costly the algorithm is to run?**
>
> We conducted our experiments on a system with the following specifications: Ubuntu 20.04 operating system, NVIDIA GeForce RTX 3090 GPU with 24GB of memory, and an Intel Xeon Gold 5218R processor. During the experiments, our model utilized approximately 8524MB of GPU memory. Notably, we found that training the model 0.5 million steps required approximately one day.
>
> ### **Q8: As the authors mention, DAAC's performance is limited in scenarios where the ground truth state is not available. Could the authors describe where the algorithm fails in partially-observed scenarios?**
>
> Due to the partially observable nature of POMDPs, there is a possibility that the expert trajectory may have identical observations at different times, leading to potential errors or ambiguities in action matching with the DA architecture. Specifically, in the maze environment, the agent may encounter similarly structured corners while navigating toward the goal. Without access to the global position, these states appear identical at the observational level, posing a risk of erroneous decision-making.

---

> > ### Comment · Reviewer_nbDi · 2023-11-22
> > **Response to authors**
> >
> > This is just to acknowledge that I read the author responses and the revised manuscript. I maintain my assessment for now and will continue to monitor the rest of the discussions.

---

### Official Review · Reviewer_pYHG · 2023-11-01

**Soundness:** 3 good
**Presentation:** 4 excellent
**Contribution:** 3 good
**Rating:** 5
**Confidence:** 4

**Summary:**

This paper proposes a new topic named imitator learning (ItorL), requiring to on-the-fly reconstruct the imitation policies based on very limited expert demonstrations for different unseen tasks. To achieve such out-of-the-box imitation capability, the authors propose a context-based imitation learning architecture Demo-Attention Actor-Critic (DAAC), which conditions on a single expert demo. The method gets good one-short imitation performance on navigation and manipulation tasks.

**Strengths:**

1.	Authors study an interesting topic since one-shot and out-of-the-box imitation learning is appealing and challenging.

2.	The proposed method is well-motivated and the designed attention architecture makes sense.

3.	The experiment results look good.

**Weaknesses:**

Please refer to the questions below.

**Questions:**

1.	The setting is unclear. Because the proposed demonstration-based attention architecture fundamentally is to retrieve a nearest neighbor state in the expert demonstration $ \tau_{\omega_{test}}$ , this method is based on an assumption that the rollout states $\tau_{agent}$ are covered by the training or context trajectories. Though it is also mentioned in Appendix F, there is no corresponding assumption in the task formulation section. I think this is an essential problem and is necessary to make an accurate formulation in Section2.1. This assumption narrows the application in more complex scenarios and makes it not a real out-of-the-box imitation method.

2.	I think the setting and proposed architecture are very similar to Mandi et al.[1]. Would you please clarify the differences and the reason why you don’t compare with it?

3.	The problem setting is very similar to some Inverse RL works (e.g., GAIL[2] and ROT[3]), i.e., requiring expert demos and online interaction. Especially, in ROT experiments, it also performs well with only a single expert demo.

[1] Mandi Z, Liu F, Lee K, et al. Towards more generalizable one-shot visual imitation learning.
[2] Ho J, Ermon S. Generative adversarial imitation learning.
[3] Haldar S, Mathur V, Yarats D, et al. Watch and match: Supercharging imitation with regularized optimal transport

---

> ### Author Response · Authors · 2023-11-20
> **Official response (part 1)**
>
> Thank you for all of the insightful questions on our paper.  We response the questions and concerns as follows:
>
> ### **Q1: The method  based on an assumption that the rollout states \tau_{\test} are covered by the training or context trajectories \tau_{agent}. The assumption in Appendix F is not no corresponding assumption in the task formulation section. I think this is an essential problem and is necessary to make an accurate formulation in Section2.1. This assumption narrows the application in more complex scenarios and makes it not a real out-of-the-box imitation method.**
>
> Thank you for your insightful observations and comments. We acknowledge the importance of a clear and accurate problem formulation at the outset and appreciate your concern regarding the assumptions made in our method.
>
> Firstly, we would like to address a potential misunderstanding related to the limitations discussed in Appendix F. **These limitations are not hidden assumptions added post-facto but are indeed consistent with the problem description provided in Definition 2.1.** We emphasize that the limitation of the DA architecture, which may compromise decision-making in a $\tau_{\Omega}-\rm{tracebackable}$ MDP set (as defined in Def. 2.1), aligns with the core scenario described in the main body of the paper. The example of states being distant from expert states for inevitable reasons just serves as an example to illustrate this point clearly.
>
> Along the way, we wish to clarify that the assumption in Def. 2.1 does not imply that the rollout states $\tau_{test}$ are necessarily covered by the training or context trajectories $\tau_{agent}$. Prop. 2.2 asks for the existence of a unified goal-conditioned policy $\beta$ (as defined in Def. 2.1). This policy $\beta$ is fundamental in ensuring that, irrespective of the unseen states that the task or unexpected environmental changes might lead the agent to, the approach to reverting to states in the demonstrations remains uniform and consistent. For instance, in navigation tasks like parking, encountering unforeseen obstacles and pedestrians is common. The demonstrations may not cover these scenarios, yet the approach to navigating these situations (avoidance followed by tracing back to the demonstration) remains consistent. This consistency is the crux of our approach. It's important to note that this consistent behavior does not apply to all states in the state space. As per Def. 2.1, focusing on the states in $\tau_{\omega}$ and $R(d_0)$ suffices to establish the feasibility of Prop. 2.2, with a full derivation and discussion presented in App. A.
>
> We recognize that our initial presentation in Section 3.2 might have inadvertently led to this misunderstanding, as the relationship between the method's limitations and Def 2.1 was not explicitly articulated. In our revised manuscript, we have made adjustments to clarify this connection, including modifications in Section 4.2 and an expanded discussion on the motivation for the beta policy in Section 2.2. We hope these revisions will dispel any misconceptions and clearly convey the applicability and robustness of our method.
>
> ### **Q2: Would you please clarify the differences between Mandi et al.[1] and DAAC and the reason why you don’t compare with it?**
>
> Thank you for your insightful question.   In our research, we consider the TRANS-BC algorithm as an approximate implementation of the method introduced by Mandi et al. [1] in scenarios involving low-dimensional state space. The model structure described in [1] utilizes a multi-head attention mechanism to extract actions and employs an additional contrastive learning loss to effectively learn representations, particularly by leveraging the similarity between adjacent frames in videos. This approach mirrors the general computational flow of a transformer, utilizing multiple attention layers to extract information from sequential inputs. However, it distinguishes itself by adapting to the input modality of video images, incorporating structures like nonlocal self-attention blocks.
>
> In contrast, our experiments are entirely state-based. Therefore, aspects of representation learning for images and other modifications tailored to high-dimensional inputs are not directly applicable to our context. Given this distinction, we deemed the TRANS-BC algorithm as a suitable proxy for comparison in our experimental setup, as it aligns more closely with the state-based nature of our research.

---

> ### Author Response · Authors · 2023-11-20
> **Official response (part 2)**
>
> ### **Q3: The problem setting is very similar to some Inverse RL works (e.g., GAIL[2] and ROT[3]), i.e., requiring expert demos and online interaction.**
>
> We understand the reviewer's concerns, and we are currently exploring inverse reinforcement learning (IRL) to assess its performance in a multi-task setting based solely on single demonstrations. The methods mentioned by the reviewer (GAIL and ROT) are suitable for single-task scenarios and we are making efforts to create a multi-task version of ROT[1]. However, currently, this modification only works effectively in a single-task settings. We intend to complete and finalize this implementation before the final paper version. Nevertheless, since this is a baseline that we have internally modified, we cannot guarantee its timely implementation during the discussion period.

---

> ### Comment · Reviewer_pYHG · 2023-11-23
>
> Thank you for the response. However, after careful consideration, I still have reservations regarding the novelty and distinction of your work in comparison to Mandi et al.
>
> 1) In architecture-wise, While your paper introduces advancements in the attention module, considering the two-year gap since the publication of Mandi et al., this modification seems incremental.
>
> 2) Your focus on state-based experiments is noted as a point of departure from Mandi et al., who used image observations. However, state-based policy learning is often viewed as less challenging than image-based learning. This perspective could potentially undermine the perceived impact of your findings.
>
> Therefore, I decide to keep my original score.

---

### Official Review · Reviewer_RXZB · 2023-11-06

**Soundness:** 1 poor
**Presentation:** 2 fair
**Contribution:** 1 poor
**Rating:** 3
**Confidence:** 3

**Summary:**

This paper proposes the problem called “Imitator learning”, where agents learn from demonstration trajectory. The paper subsequently proposes “DAAC”, reweighting actions from the demonstration trajectory.

**Strengths:**

- The paper is well motivated

**Weaknesses:**

- I am not sure whether the newly proposed “Imitator learning” is unique. It seems to me that the problem setup is within the definition of few-shot imitation learning or meta learning.
- Writing: the definitions and propositions in section 2.2 do not add value. They do not offer insights and are purely definitions. It seems to be that the entire section 2.2 can be replaced with one sentence: “we train a goal-conditioned policy”.
- The proposed method is only tested on a simple maze-like environment. I would like to see evaluations on more mainstream control benchmarks, such as DMC [1] or robosuite [2].
- More baselines from IL methods. Such as GAIL [3] or SQIL [4].

[1] DeepMind Control Suite, Tassa et al.

[2] Robosuite, Zhu et al.

[3] Generative Adversarial Imitation Learning, Ho et al.

[4] SQIL: Imitation Learning via Reinforcement Learning with Sparse Rewards, Reddy et al.

**Questions:**

Same as the concerns raised in the weaknesses section.

---

> ### Author Response · Authors · 2023-11-20
> **Offcial response (part 1)**
>
> Thank you for your insightful comments. we response the questions and concerns as follows:
> ### **Q1: It seems to me that the problem setup is within the definition of few-shot imitation learning or meta learning.**
>
> While it may initially appear akin to few-shot imitation, there are pivotal differences, which we elaborated in Sections 2.1 and 4, and Appendix C of our paper. Below is a concise summary to address your specific concerns:
>
> 1. **Distinction from Few-shot Meta-Imitation Learning (Meta-IL)**:
>     - Traditional few-shot Meta-IL, as you noted, strives to develop a generalizable policy capable of adapting to new tasks with minimal expert trajectories, primarily through model-agnostic meta-learning (MAML) approaches [1]. These methods necessitate online interactions and additional computational resources for gradient updates and calibration before deployment.
>     - In contrast, ItorL constructs an imitator policy, $\Pi(a∣s, \tau)$, exclusively conditioned on a expert demonstration, **without the need for any fine-tuning** during deployment. This key distinction eliminates the requirement for post-training adjustments, setting ItorL apart from the conventional few-shot Meta-IL paradigm.
> 2. **Differentiation from One-shot Meta-IL**:
>     - One-shot Meta-IL typically relies on context-based policy models, like Transformers, to process demonstrations [2]. Although powerful in fitting, these approaches often struggle with prediction errors in unseen states, compounded by the inherent limitations of Behavioral Cloning (BC) in generalizing to new tasks.
>     - ItorL, on the other hand, permits interaction with simulators during training and assumes expert-level quality in the demonstration set. This not only allows ItorL to more accurately imitate expert behavior but also to learn generalized responses to situations not covered in the demonstration set. The focus on expert demonstrations and the allowance for simulator interactions fundamentally set ItorL apart, enabling **enhanced imitation learning with fewer demonstrations** compared to other methodologies.
>
> We appreciate your concern regarding the potential confusion due to the major discussions being in the appendix. In response, we have revised the related work section for better clarity and accessibility. Please refer to Section 2 in the revised manuscript for a detailed exposition.
>
> ### **Q2: the definitions and propositions in section 2.2 do not add value. They do not offer insights and are purely definitions. It seems to be that the entire section 2.2 can be replaced with one sentence: “we train a goal-conditioned policy”.**
>
> Thank you for your valuable feedback on Section 2.2 of our paper. We respectfully disagree with the suggestion to condense this section into a single sentence, and here's why:
>
> 1. **Rigorous Problem Formulation**: Section 2.2 is pivotal as it provides a rigorous formulation of the specific problems that our Imitator learning (ItoL) approach can address, particularly when based on a single demonstration. In academic research, it's crucial to define the solution scope clearly. This section is not merely about empirical implementations; it's about setting a theoretical foundation for the application of ItoL.
> 2. **Misinterpretation of Proposition 2.2**: The proposition in question does not imply that we train a goal-conditioned policy. Rather, it posits that **if** a unified goal-conditioned policy exists for a given task set, then there must be at least one corresponding unified imitator policy. This is a theoretical assertion about the nature of the task set and the potential existence of an imitator policy, rather than a direct description of our method's implementation.
>
> Acknowledging your feedback, we have revised Section 2.2 to better articulate the motivation behind the definition of the goal-conditioned policy *β*. This revision aims to clarify any misconceptions and emphasize the theoretical underpinnings that distinguish our approach from conventional methods.

---

> ### Author Response · Authors · 2023-11-20
> **Official resposne (part 2)**
>
> ### **Q3: The proposed method is only tested on a simple maze-like environment.  I would like to see evaluations on more mainstream control benchmarks, such as DMC [1] or robosuite [2].**
>
> Thank you for your valuable feedback. As outlined in Section 5.1, the maze environments, while seemingly straightforward, encompass challenging tasks that are far from trivial.
>
> 1. **Limited State Space Perception**: The state space in our tasks is constrained to a local view, defined by eight rays less than five steps in length. This limitation imposes a non-trivial complexity to the navigation and decision-making processes within the environment.
> 2. **Unpredictable Obstacle Dynamics**: Unlike the demonstrations, our environment includes obstacles that randomly appear, adding an element of unpredictability and requiring adaptive strategies for successful navigation.
> 3. **Random Map Generation and Single Trajectory Imitation**: The tasks involve navigating through randomly generated maps. Moreover, the imitation learning is accomplished using only a single trajectory, which significantly elevates the task difficulty.
>
> To our knowledge, there are no existing methods capable of effectively addressing these specific conditions. This uniqueness underscores the innovative aspect of our approach. Since the maze environment is efficient for rollout and easy to construct a number of test cases via generate different map, we defend that it is a better benchmark for ItorL compared with existed robotics benchmark.
>
> Beyond the maze setting, we have conducted extensive validations of our method in a variety of robotic environments. This includes well-known Gym environments such as ‘pusher’, ‘reacher’ and a mixture of manipulation tasks, demonstrating the method's adaptability and effectiveness, which should not be ignored.
>
> Finally, addressing your request, we deliberately refrained from adjusting any parameters and transferred our algorithm to the pick-and-place environment in Meta-World. The final success rates for each method at different timesteps are as follows:
>
> | Time step | DAAC | DCRL | CbMRL | TRANS-BC |
> | --- | --- | --- | --- | --- |
> | 0.5M | 0.980 | 0.240 | 0.640 | 0.160 |
> | 1.5M |  1.00 | 0.925 | 1.00 | 0.084 |
>
> The results demonstrate strong performance, surpassing all baseline algorithms. Note that our method converges at 0.5 million timesteps. While DCRL and CbMRL finally complete the task in later stages, it takes them 1.5 million timesteps, resulting in **three times the computational effort** compared to our approach. We believe this showcases the robustness and effectiveness of our approach without extensive parameter adjustments.
>
> These results not only demonstrate our method's efficacy in the initial maze-like environment but also its superior performance in mainstream and more complex control benchmarks. We believe these validations effectively address your concerns and underscore the robustness and versatility of our approach.
>
> ### **Q4: More baselines from IL methods. Such as GAIL [3] or SQIL [4].**
>
> Thanks for the suggestions and We understand the reviewer's concerns. The IL methods mentioned by the reviewer, GAIL, SQIL and also ROT [1] mentioned by pYHG, are suitable for single-task scenarios and cannot be directly used as baselines for comparison. We have made efforts to construct a multi-task version of the IL algorithms based on the official implementations; however, currently, this modification only works effectively in single-task settings. We intend to complete and finalize this implementation before the final paper version. Nevertheless, since this is a baseline that we have internally modified, we cannot guarantee its timely implementation during the discussion period.
>
> [1]  Haldar S, Mathur V, Yarats D, et al. Watch and match: Supercharging imitation with regularized optimal transport.

---

### Author Response · Authors · 2023-11-20
**general response**

We express our sincere gratitude for your constructive feedback, insightful remarks, and the recognition of the strengths in our work. Your observations have been invaluable in refining our manuscript.

We are encouraged by the positive responses, notably:

- Clear and Well-Structured Presentation: The paper has been recognized for its clarity and well-organized structure, effectively presenting the main ideas. (Reviewer pYHG, 7pta)

- Innovative Concept of Imitator Learning (ItorL): Reviewers appreciated the introduction of ItorL, emphasizing its relevance and practical application in learning from limited expert demonstrations across various unseen tasks. (Reviewer pYHG, nbDi, 7pta)

- Effectiveness of Demo-Attention Actor-Critic (DAAC) Approach: The DAAC method has been commended for its effective solution to the ItorL problem, demonstrating superior performance in comparison to other methods in meta-RL. (Reviewer nbDi, 7pta)

- Strong Empirical Results: The experiments conducted were found to be sufficient and convincing, reflecting the advantages of the proposed method.  (Reviewer 7pta)


The feedback has guided significant improvements in our manuscript:

- We enhanced the Related Work section to clarify the differences and similarities with related IL concepts, including meta-RL and meta-imitation learning. This discussion is now prominently featured in Section 3.
- We polished Section 2.2 to further demonstrate the motivation behind Proposition 2.2 and how practical the assumptions are. We also give several example that the assumptions hold for the readers to understand our motivation.
- We polished Section 4.2 to clarify how the DA architecture works as a regularizer, and the relationship between assumptions in Proposition 2.2 and the claimed limitations in faraway states.
- We provided more experiments on Meta-world (See response below) to verify that DAAC’s is a universal algorithm in different tasks.
- We provided more visualisations on trajectories learned by different imitation paradigms (Appendix L) to illustrate the difference of the imitation policies learned by different paradigm, including meta imitation learning and meta reinforcement learning.

All the revisions that are newly added are marked with blue, while the contents that are moved from another place are colored purple.

We appreciate the opportunity to discuss our work and are committed to continuing this dialogue to further advance the field. Your feedback has been instrumental in elevating the quality of our research.

---

### Author Response · Authors · 2023-11-23
**additional experimental results**

The authors thank all reviewers once again for their valuable feedback. We have noted that the reviewers are interested in general benchmark tests. In response to this, we have conducted additional experiments on MetaWorld. Previously, we supplemented results on the Pick Place environment within MetaWorld, where our method demonstrated excellent performance and convergence speed. Given that some algorithms can also complete this task in later stages, we transfer these methods to a more complex multi-task learning scenario to highlight performance differences, including pick-and-place, pick-place-wall, and pick-out-of-hole within MetaWorld. The results demonstrate that in multi-task scenarios, our algorithm is the only one that successfully learns to finish the task.

| Time step | DAAC | DCRL | CbMRL | TRANS-BC |
| --- | --- | --- | --- | --- |
| 0.5 M | 0.28 | 0.02 | 0.02 | 0.02 |

---

### Meta-Review · Area_Chair_PUWJ · 2023-12-06

**Metareview:**

(a) Summary

The paper introduces "Imitator Learning" (ItorL), a purportedly novel concept in imitation learning that focuses on out-of-the-box imitation from limited demonstrations.  The authors propose the Demo-Attention Actor-Critic (DAAC) method, integrating imitation learning into a reinforcement learning framework. DAAC uses a demonstration-based attention architecture to adaptively output actions by attending to relevant states in demonstrations. The authors claim that DAAC outperforms existing imitation methods in both seen and unseen tasks, supported by experiments on a new navigation benchmark and a robot environment.

(b) Strengths
(+) Thorough Experimental Evaluation (Reviewers pYHG, nbDi, 7pta): The experiments conducted are extensive reflecting the advantages of the proposed method. The additional experiments on MetaWorld provide further evidence of the method's effectiveness and versatility.


(+) Clear and Structured Presentation (Reviewer pYHG): The paper is recognized for its clarity and well-organized structure, effectively presenting the main ideas and methodologies.

(+) Effective DAAC Approach (Reviewers nbDi, 7pta): The DAAC method is a reasonable and effective solution to the proposed ItorL problem, and outperforms other methods in meta-RL

(c) Weaknesses of the Paper:

(-) Problem Formulation Novelty (Reviewer RXZB): The novelty of the problem formulation is questioned, as it appears to be a variant of few-shot imitation learning. The distinction between fine-tuning vs. conditioning on the demonstration seems more a detail of policy architecture than a conceptual innovation.

(-) Limited Theoretical Insight (Reviewer RXZB): The definitions and propositions in section 2.2 are seen as adding little value and not offering deep insights.

(-) Generalization Concerns (Reviewer pYHG): The paper's methodology relies on the assumption that rollout states are covered by training or context trajectories, which may limit its application in more complex scenarios and challenge its out-of-the-box imitation capability.

(-) Comparison with Existing Work (Reviewer pYHG, nbDi): There is a need for more elaboration on the differences between ItorL and existing methods, such as those in inverse RL and meta-RL. Also, the paper could benefit from additional comparisons with related works.

**Justification For Why Not Higher Score:**

The "imitator learning" problem appears to essentially be few-shot imitation learning. Whether the policy is fine-tuning vs conditioning on the demonstration appears to be a detail of the policy architecture. There is no real theoretical analysis of the approach. The experimental evaluation, even with the addition of Meta world, isn't signifiant enough to cross the bar for an accept.

**Justification For Why Not Lower Score:**

N/A

---

### Decision · Program_Chairs · 2024-01-16

Reject